# Mining multi-center heterogeneous medical data with distributed synthetic learning

Qi Chang [1,9], Zhennan Yan [2,9], Mu Zhou[2,3,9], Hui Qu[1], Xiaoxiao He[1], Han Zhang[1], Lohendran Baskaran[4], Subhi Al'Aref[5], Hongsheng Li[6,7,10] ✉, Shaoting Zhang[3,7,8,10] ✉ & Dimitris N. Metaxas [1,10] ✉

Overcoming barriers on the use of multi-center data for medical analytics is challenging due to privacy protection and data heterogeneity in the healthcare system. In this study, we propose the Distributed Synthetic Learning (DSL) architecture to learn across multiple medical centers and ensure the protection of sensitive personal information. DSL enables the building of a homogeneous dataset with entirely synthetic medical images via a form of GAN-based synthetic learning. The proposed DSL architecture has the following key functionalities: multi-modality learning, missing modality completion learning, and continual learning. We systematically evaluate the performance of DSL on different medical applications using cardiac computed tomography angiography (CTA), brain tumor MRI, and histopathology nuclei datasets. Extensive experiments demonstrate the superior performance of DSL as a high-quality synthetic medical image provider by the use of an ideal synthetic quality metric called Dist-FID. We show that DSL can be adapted to heterogeneous data and remarkably outperforms the real misaligned modalities segmentation model by 55% and the temporal datasets segmentation model by 8%.

Multi-center healthcare data sharing is challenging due to privacy regulations and heterogeneity to advance medical research. It is widely known that sufficient medical data samples are the foundation to enable successful machine learning algorithms[1] in a range of medical fields such as neuroscience[2], genetics[3], drug discovery[4,5], and disease diagnosis and prognosis[6–8]. Deep learning utilizes complex models with millions or billions of parameters and requires significantly larger data samples compared to classical machine learning algorithms to achieve its remarkable performance. However, the available medical datasets for machine learning research often lack the necessary data size with sufficient annotations. The representative national lung screening CT cohort includes 54 thousand[9] CT images, accounting for only 0.06% of the annual acquired images in the United States[10]. Such a centralized dataset is significantly smaller in size than the vision benchmark ImageNet[11] with 14 million images.

Although cross-institutional collaboration could accelerate medical AI research, current data sharing and aggregation are challenged by the mandatory privacy protection of patient records and related review mechanisms. The Institutional Review Board (IRB)[12] which provides comprehensive guidelines to protect a subject's privacy, and a series of data protection regulations and protocols, including HIPAA[13,14], and EU GDPR[15,16], are necessary to protect and use patient data, but at the same time create daunting bottlenecks for sharing real-world data across institutions. In addition, collecting data in a

[1]Department of Computer Science, Rutgers University, Piscataway, NJ, USA. [2]SenseBrain Research, Princeton, NJ, USA. [3]Shanghai Artificial Intelligence Laboratory, Shanghai, China. [4]Department of Cardiovascular Medicine, National Heart Centre Singapore, and Duke-National University Of Singapore, Singapore, Singapore. [5]Department of Medicine, Division of Cardiology, University of Arkansas for Medical Sciences, Little Rock, AR, USA. [6]Chinese University of Hong Kong, Hong Kong SAR, China. [7]Centre for Perceptual and Interactive Intelligence (CPII), Hong Kong SAR, China. [8]SenseTime, Shanghai, China. [9]These authors contributed equally: Qi Chang, Zhennan Yan, Mu Zhou. [10]These authors jointly supervised this work: Hongsheng Li, Shaoting Zhang, Dimitris N. Metaxas. ✉e-mail: hsli@ee.cuhk.edu.hk; zhangshaoting@pjlab.org.cn; dnm@cs.rutgers.edu

centralized hub increases the risk of information leakage as routine data anonymization can not guarantee data privacy protection[17]. ue to the above issues, synthetic data generation and processing methods are gaining momentum since they have the potential to provide large-scale accessible data without compromising privacy[18]. Synthetic data have shown their potential to complement real-world data and facilitate all stages of model training[19]. For instance, training generative methods such as generative adversarial networks can generate high-quality synthetic data to boost the performance of image reconstruction[20], classification[21], segmentation[22], and domain translation[23].

Multi-center medical records are inherently heterogeneous due to the various hospital acquisition protocols, scanner types, data modalities, and patient outcomes[24,25]. With the growth of medical data volumes, integrative analysis of multi-center data is becoming increasingly important to accelerate clinical workflow quality and quantitation. Federated learning (FL)[26–28] methods learn a base model across multiple data centers. However, current FL approaches are often inefficient in learning heterogeneous data[29] and require retraining to use improved future deep-learning techniques[30]. In addition, in clinical centers, the accessibility and availability of data depend on IRB restrictions which is a clear hurdle to support continual learning in real-world scenarios[31].

In this study, we propose a novel distributed synthetic learning (DSL) architecture to address patient privacy and data heterogeneity in using multi-center data. Our distributed synthetic learning architecture can generate high-quality synthetic images from various multi-center image modalities and also supports continual learning. We demonstrate the privacy-protection advantages of our approach experimentally by utilizing the generated synthetic images only in the downstream segmentation and classification models which we then test on real-world patient data. We then conduct extensive evaluations of DSL on three different multi-center medical data sets, i.e., CTA cardiac data, multi-modality MRI brain data, and histopathology data. The data from each of the centers have been acquired with varying protocols, scanners, microscopes, and settings, which makes them highly heterogeneous. The results demonstrate the improved performance of DSL compared to the state of the art for privacy-preserving learning. Finally, we measure the synthetic image quality using the proposed distributed metric termed Dist-FID. We believe Dist-FID will be the replacement of FID[32] when distributed learning is used for medical image generation.

## Results

In this section, we conduct experiments to systematically and quantitatively validate DSL in a variety of settings. We investigate the effectiveness of DSL in image segmentation tasks using multi-source heterogeneous data from distributed data centers. Then, we explore two practical cases involving multi-modality and missing-modality image data, emphasizing the DSL capability of learning the misaligned data distribution. Finally, we validate DSL in a challenging continual learning experiment where the participating data centers may enter and exit the deep-learning process at various time points.

### Overview of approach

As shown in Fig. 1, DSL is designed to address the medical privacy, data heterogeneity, and multi-modality challenges in multi-center datasets. Figure 1a illustrates that DSL is comprised of one central generator and multiple distributed discriminators located in different data centers (nodes). The central generator takes task-specific inputs (e.g., segmentation masks) and generates synthetic images that follow a similar distribution as the data distribution of the data centers. Using distributed adversarial learning, our architecture ensures that the central generator learns a joint data distribution without direct access to the private data of the data centers. The

learned generator produces a large public synthetic database for use in downstream tasks. Our method can not only learn from single-modality heterogeneous medical data but also from multi-modality data shown in Fig. 1b. DSL can handle a complex and heterogeneous situation even when some modalities are missing as shown in Fig. 1c. This is achieved because the missing-modality completion of DSL's central generator learns to synthesize the missing modality. Furthermore, DSL can be applied in a challenging continual learning setting where the data centers are only temporarily available as shown in Fig. 1d. The continual-learning ability of DSL keeps learning from multiple data centers and prevents the generator model from catastrophic forgetting[31]. In particular, we use continual learning in a four-separate-data-center setting, where the data centers are accessed sequentially. In the following, we provide details on the performance of DSL and comparisons to three state-of-the-art methods, namely, federated-learning GAN (FLGAN)[33], AsynDGAN[30], and FedMed-GAN[23]. FLGAN represents the typical way of federated learning by training and aggregating the same model across centers. AsynDGAN is a baseline distributed GAN model that showed promising performance in learning to synthesize T2 brain images and histopathology images. It shares a similar architecture as DSL, but lacks support for heterogeneous data and an efficient model selection strategy. FedMed-GAN is a recently proposed federated-learning GAN method that shows state-of-the-art performance for cross-modality brain image synthesis. In our implementations, all comparing GANs used the same backbone network. The datasets used in all experiments are summarized in Table 1. The training and testing samples are split at the patient level. The training images are used to learn the DSL models and other comparing methods. The testing images are used in the downstream tasks for evaluation.

### Distributed synthetic learning

We perform several experiments with different types of multi-center datasets. First, we demonstrate that DSL can learn from multi-center heterogeneous cardiac CTA images to obtain a central generator that can generate high-fidelity synthetic CTA images by using masked cardiac images as input. Masking is done to outline the key components of the heart such as the ventricles, the atria, and the aorta. We use three public cardiac CTA datasets, WHS, CAT08, and ASOCA, acquired from multiple institutes globally (see Data collection and processing for details) for evaluation.

Figure 2c and Table 2 show the segmentation performance of different methods learned from the heterogeneous cardiac multi-center datasets. In Table 2, the first row of Real-All shows the segmentation results based on centralized learning of all three cardiac datasets together, which we use as a baseline to compare our method. Due to the data restrictions outlined previously, this is a hypothetical learning scenario. A possible real scenario is to learn segmentation separately on data from a single data center. The experiments corresponding to Real-WHS, Real-CAT08, and Real-ASOCA show segmentation results using the same segmentation method and data from a single data center. The rows of FLGAN, AsynDGAN, FedMed-GAN, and DSL correspond to experiments that segmentation models are trained on synthetic data, which is generated by different generative learning models. They use the three private cardiac datasets from the above data centers. We observe that the segmentation model relying on a single dataset shows significantly inferior performance since the Dice score of Real-CAT08 is about 25% less than the Real-All result. In terms of the Dice score, 95% Hausdorff Distance (HD95), and average Surface Distance (SD), DSL demonstrates superior performance compared with other state-of-the-art federated learning methods or direct learning methods that use a single dataset. For example, the Dice score of FLGAN with federated averaging aggregation is 0.709, which is significantly lower ($p = 0.0001$ by a two-sided paired $t$-test) than DSL with Dice of 0.864.

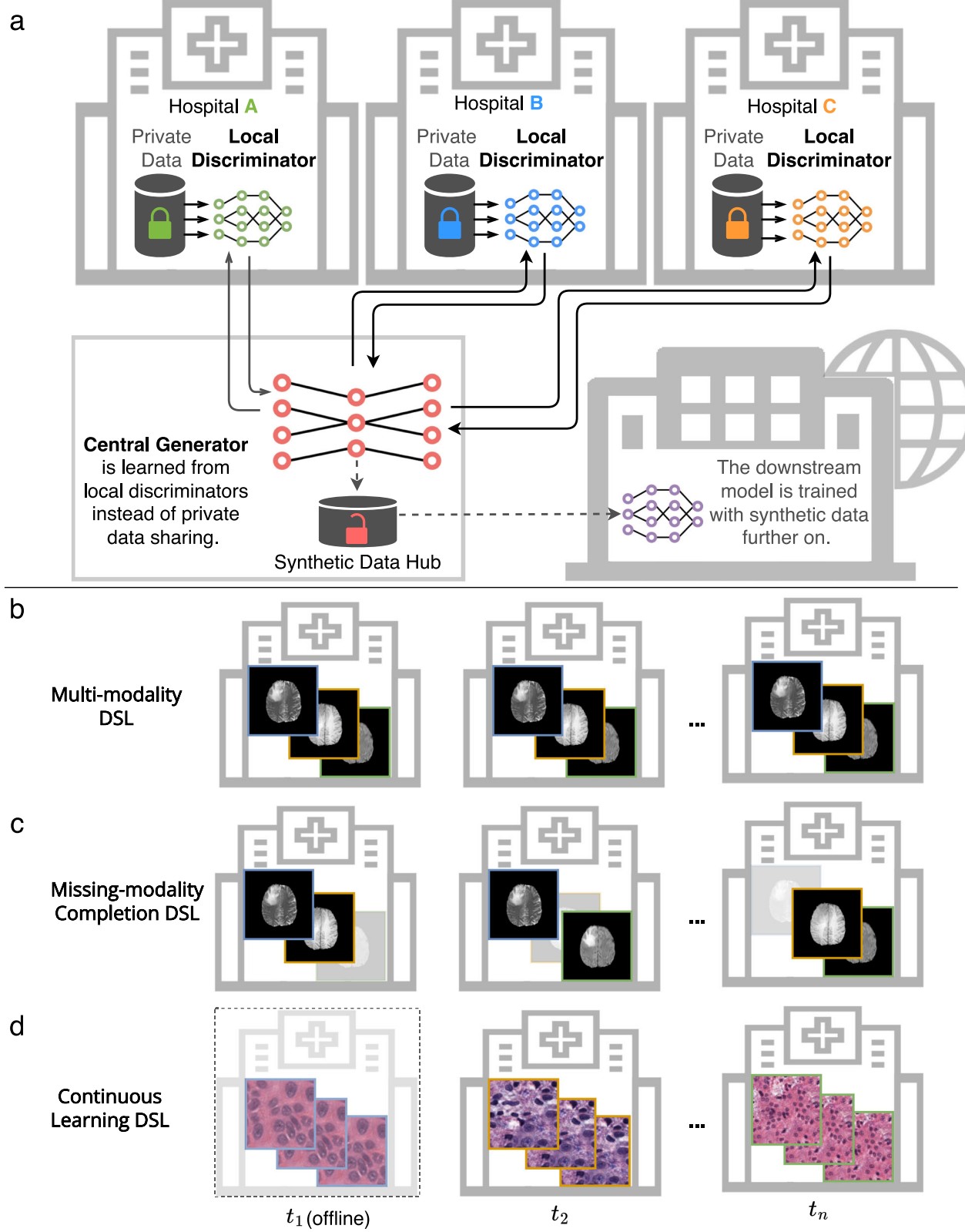

**Fig. 1 | Overview of the DSL architecture. a** The architecture contains one central generator and multiple distributed discriminators, each located in a medical entity. Then the well-trained generator can be used as an image provider to build a synthetic database for downstream machine-learning tasks. The following three rows show three different scenarios for heterogeneous medical data. In (**b**), the generator learns and generates multi-modality synthetic images at the same time. In (**c**), the data centers provide data with misaligned modalities. To highlight, the proposed DSL framework could leverage the misaligned information from multiple data centers and synthesize unified multi-modality images. In (**d**), the temporal data centers can only be accessed sequentially. DSL can learn from each temporal dataset without catastrophically forgetting what the model has previously learned.

**Table 1 | Summary of datasets**

| Cardiac CTA | WHS[64–66] | ASOCA[67,68] | CAT08[69] | – |
|---|---|---|---|---|
| Train Subjects (Image#) | 20 (3031) | 32 (4642) | 26 (3568) | – |
| Test Subjects (Image#) | 40 (6360) | 8 (1180) | 6 (756) | – |
| Avg Spacing (mm³) | $0.44^2 \times 0.6$ | $0.4^2 \times 0.625$ | $0.32^2 \times 0.4$ | – |
| Scanner | Philips | Unknown | Siemens | – |
| **BraTS18[72–74]** | **CBICA** | **TCIA** | **OTHER** | **–** |
| Train Subjects (Image#) | 69 (4638) | 85 (5736) | 14 (975) | – |
| Test Subjects (Image#) | 19 (1165) | 17 (1172) | 6 (393) | – |
| **Nuclei[75] (tasks#)** | **Liver (t1)** | **Breast (t2)** | **Kidney (t3)** | **Prostate (t4)** |
| Train Subjects (Nuclei#) | 4 (1906) | 4 (1508) | 4 (4866) | 4 (1634) |
| Test Subjects (Nuclei#) | 2 (838) | 2 (707) | 2 (716) | 2 (766) |

The cardiac CTA data were collected from three different sources and images were acquired from different devices with various spacings. The BraTS18 data was collected from different data centers with four modalities. In the missing modality completion setting, the modalities of different centers are misaligned. In the Nuclei dataset, each organ has a different data size and we conduct continual learning by using them sequentially.

We also assess the quality of synthetic images and Fig. 2a shows examples of synthetic images of each generative method for the heterogeneous cardiac datasets. By selecting the best model with the lowest Dist-FID score to generate synthetic data, DSL outperforms the other federated and distributed GANs in terms of peak signal-to-noise ratio (PSNR) and structural similarity index measure (SSIM). The primary reason for using Dist-FID in model selection is that such metric can reflect the actual image quality of all distributed datasets, rather than calculating FID by using a subset of all datasets or one of the private datasets. From the curves in Fig. 2b, we recognize that Dist-FID is more favorable than other image quality measurement strategies to select the optimized epoch model, and Dist-FID can be an ideal replacement for the Frechet Inception Distance (FID)[32] in the distributed system. For instance, the synthetic images by FLGAN[33], have a Dist-FID score of 72.05, PSNR of 13.55, and SSIM of 0.414, while our DSL can generate better-quality images with a Dist-FID of 61.09, PSNR of 16.04, and SSIM 0.456.

**Multi-modality distributed synthetic learning**

We further validate our approach through multi-modality distributed synthetic learning (MM-DSL) on the multi-modality MRI datasets. We simulate three data centers, including CBICA, TCIA, and OTHER. Taking the three tumor sub-region labels and the brain skull as the input, MM-DSL learns to generate realistic multi-modality (T1, T1c, T2, Flair) brain MRI images (see Data collection and processing for details). For the downstream machine-learning task, we focus on the whole-tumor segmentation.

From Table 3 and Fig. 3, MM-DSL achieves higher performance than the baselines of FLGAN, FedMed-GAN, and AsynDGAN on multi-modality image generation and whole-tumor segmentation on the BraTS dataset. The Dice score of MM-DSL is 0.829, which is remarkably higher than FLGAN with 0.736, FedMed-GAN with 0.73, and AsynDGAN with 0.802. However, the models learned from a single dataset may be unstable due to the unbalanced size and different image quality. For instance, the Dice score of Real-OTHER, 0.765, is significantly lower than Real-TCIA with 0.823. In addition, the synthetic data generated by MM-DSL could also be treated as a data augmentation to boost the performance of the single real image dataset. For instance, by combining MM-DSL's synthetic data and Real-TCIA data, the Syn+Real-TCIA achieves a Dice score of 0.854, which is better than Real-TCIA with 0.823 and MM-DSL alone with 0.829. It is worth mentioning that the improvement achieved by this type of data augmentation can be more remarkable for smaller data size from a single data center. For example, the segmentation Dice score of the Real-OTHER dataset with 0.765 is boosted with augmentation by 7.7% to 0.824.

To validate the generalization ability of DSL, we further verify the MM-DSL synthetic image quality with another downstream binary classification task of image-based tumor region recognition. We split the BraTS dataset into two categories (positive 11,349 and negative 14,691) to train the classifiers and test on 6510 real-world brain MRI images. As reported in Supplementary Table 1, different methods used either real samples or synthetic samples in the training, and all used the same VGG network architecture[34] with BCE loss. The accuracy of MM-DSL is 0.897, which is significantly higher than the other synthetic-image-based methods. For instance, MM-DSL outperforms FLGAN by 27%. MM-DSL also achieves better performance than the segmentation model trained by using a small-scale real dataset (Real-OTHER), which only obtains an accuracy of 0.74%.

**Missing-modality completion distributed synthetic learning**

We evaluate the robustness of our method in a complex and heterogeneous setting where the MRI modalities are misaligned across data centers. In particular, we test the robustness by removing a different modality from each of the three data centers. As a result, we removed all Flair images from the CBICA center, the T1c images from the TCIA center, and the T2 images from the OTHER center. This is a challenging scenario as the data modalities are different across the data centers. To be able to learn to generate complete-modality synthetic data in this challenging scenario, we adapt the number of discriminators in the MM-DSL architecture, noted as Hetero-MM-DSL (see Methods for details). We compare Hetero-MM-DSL with the federated segmentation method FedSeg (https://fedsegment.github.io/home), which learns directly from multi-channel real data. Real-FedSeg represents the FedSeg when it learns from complete-modality real data, while Hetero-Real-FedSeg is FedSeg when it learns from missing-modality data. Each missing modality was represented as a channel of all zeros. We selected FedSeg to compare with our DSL for the following two major reasons. First, FedSeg is a federated learning method that can learn from distributed data. Second, in this case, directly learning a segmentation model from real data is considered an upper bound of learning from synthetic data. Since the MM-DSL already achieved better results than the other GAN-based methods as shown in Tables 2 and 3, we just need to compare the results of Hetero-MM-DSL with the results of Hetero-Real-FedSeg and present them in Table 4.

From Table 4 and Fig. 4, Hetero-MM-DSL achieves better segmentation performance on missing modality settings, while the federated-learning method significantly underperforms. For example, the Dice score of Hetero-MM-DSL is 0.795, which is significantly higher than the 0.353 Dice score of Hetero-Real-FedSeg. In addition, Hetero-MM-DSL can handle this challenging problem with a small performance loss of 4% (0.795 Dice score) compared to MM-DSL (0.829 Dice score) and achieved this by completing the missing modalities with synthetic images. In contrast, the performance of FedSeg, which uses the FedAvg optimization method, is significantly reduced due to the

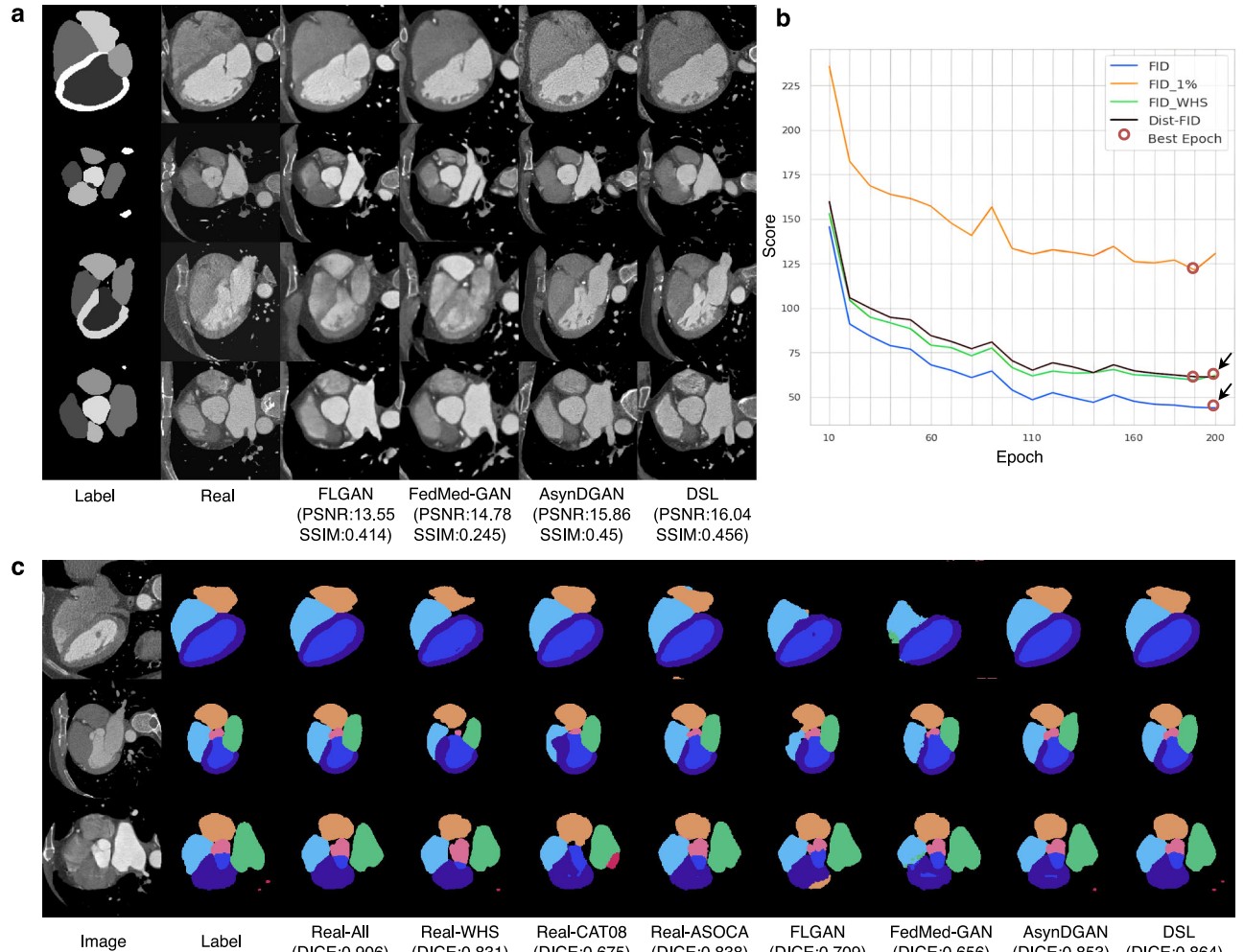

**Fig. 2 | Qualitative results of the CTA heart experiment. a** Four image generation examples for the GAN-based methods. Each row has the multi-component mask image of the heart (Label), which is the input of the image generation, the corresponding real CTA image (Real), and the synthetic images generated by different methods (FLGAN, FedMed-GAN, AsynDGAN, DSL). DSL generates images with higher scores in terms of peak signal-to-noise ratio (PSNR) and structural similarity index measure (SSIM). **b** The Dist-FID and FID score curves over the training epochs. FID is calculated using real data from all data centers. The red circles indicate the best epoch for each method, while the arrows show the consistency of FID and Dist-FID scores. **c** Three examples of segmentation results for different methods vs the ground truth Label. The segmentation model learned from DSL's synthetic data obtains more accurate results than the other methods (Real-WHS, Real-CAT08, Real-ASOCA, FLGAN, FedMed-GAN, AsynDGAN) and is comparable to centralized learning (Real-All).

**Table 2 | Quantitative results of cardiac CTA segmentation**

| Data/Method | Average Dice↑ | Average HD95 (mm)↓ | Average SD (mm)↓ |
|---|---|---|---|
| Real-All | 0.906 ± 0.037 | 6.89 ± 5.69 | 1.67 ± 0.97 |
| Real-WHS | 0.831 ± 0.086 | 13.81 ± 9.34 | 3.37 ± 2.15 |
| Real-CAT08 | 0.675 ± 0.145 | 33.24 ± 12.12 | 9.28 ± 4.80 |
| Real-ASOCA | 0.838 ± 0.093 | 18.86 ± 14.47 | 3.94 ± 2.87 |
| FLGAN | 0.709 ± 0.153 | 37.33 ± 15.47 | 6.14 ± 2.89 |
| FedMed-GAN | 0.656 ± 0.199 | 45.68 ± 10.97 | 8.45 ± 4.14 |
| AsynDGAN | 0.853 ± 0.067 | 20.58 ± 10.28 | 3.16 ± 1.87 |
| DSL | 0.864 ± 0.068 | 13.23 ± 7.93 | 2.85 ± 1.66 |

The reported results are the average score of seven sub-structures (mean ± standard deviation) in terms of Dice score, 95% quantile of Hausdorff distance (HD95), and average surface distance (SD). In the first column, 'Real-' indicates the segmentation model is trained from original real images, otherwise the model is trained from synthetic images. Real-All is the result of centralized learning from all three datasets together, which is an ideal setting. The rows of Real-WHS, Real-CAT08, and Real-ASOCA show the segmentation results by learning from a single data source, respectively. The FLGAN, AsynDGAN, FedMed-GAN, and our DSL are different generative models learning from the three distributed datasets. DSL outperforms the other generative approaches and the models learned from real data of a single data center.

different missing data modalities from each data center. For example, the Dice score of Real-FedSeg, which is 0.839 dropped significantly by 58% to 0.353 when Hetero-Real-FedSeg is used.

## Continual synthetic learning

We also evaluate our approach in a continual synthetic learning scenario with temporal datasets from multiple data centers, each consisting of data from a different organ, namely the liver, breast, kidney, and prostate. To simulate time, each dataset is exclusively accessed for a defined period of time during the learning process, starting from the liver dataset. Thus, there are four sequential continual learning tasks that use the liver, breast, kidney, and prostate datasets, respectively.

Our approach is based on the modification of DSL by incorporating a reminding loss (described in "Methods") for continual learning, which we term CL-DSL. We compare the proposed CL-DSL with two baseline approaches and one state-of-the-art method, TDGAN[35]. All four methods learn generative models, use the same network architecture for the generator and discriminator, while the main difference among them is the loss functions. The first baseline approach we term Joint Continual Learning (JCL), does not include distributed learning of a generative model. It leans a GAN model in a centralized sequential

**Table 3 | Quantitative results of whole tumor segmentation**

| Data/Method | Dice ↑ | HD95 (mm)↓ | SD (mm)↓ |
|---|---|---|---|
| Real-All | 0.862 ± 0.128 | 7.56 ± 10.90 | 1.48 ± 2.02 |
| Real-CBICA | 0.801 ± 0.142 | 27.11 ± 25.12 | 4.54 ± 4.42 |
| Real-TCIA | 0.823 ± 0.117 | 10.32 ± 10.53 | 1.90 ± 1.19 |
| Real-OTHER | 0.765 ± 0.167 | 15.61 ± 13.39 | 2.81 ± 2.36 |
| FLGAN | 0.736 ± 0.197 | 19.23 ± 17.19 | 3.50 ± 2.69 |
| FedMed-GAN | 0.730 ± 0.213 | 18.40 ± 17.17 | 3.49 ± 2.93 |
| AsynDGAN | 0.802 ± 0.162 | 15.95 ± 15.05 | 2.43 ± 1.66 |
| MM-DSL | 0.829 ± 0.128 | 11.50 ± 12.82 | 1.99 ± 1.86 |
| Syn + Real-CBICA | 0.841 ± 0.156 | 9.56 ± 13.37 | 1.91 ± 3.07 |
| Syn + Real-TCIA | 0.854 ± 0.093 | 10.50 ± 12.29 | 1.71 ± 1.30 |
| Syn + Real-OTHER | 0.824 ± 0.126 | 17.83 ± 24.08 | 2.81 ± 3.46 |

All methods learn from multi-modality brain MRI data. The reported results (mean ± standard deviation) include Dice score, 95% quantile of Hausdorff distance (HD95), and average surface distance (SD). In the first column, 'Real-' indicates the model trained from original real images, otherwise, the model is trained from synthetic images. Real-All merges together all data of CBICA, TCIA, and OTHER. 'Syn + Real-' represents synthetic data augmentation by adding the synthetic data from MM-DSL. MM-DSL outperforms the other generative approaches and the models learned from real data of a single data center. MM-DSL's synthetic data can also improve the segmentation model learned from a single data center by the 'Syn+Real' augmentation.

way by accumulating all the data in one location for learning. In particular, after learning task 1, it keeps the copy of task 1 data and adds task 2 data into the training set to emulate continual training of the GAN. In contrast, the other methods conduct the continual learning by keeping the learned knowledge in the model, while the datasets are in different data centers. The second baseline we term Fine-Tuning, uses the same learning architecture as CL-DSL, but without the reminding loss. It only keeps fine-tuning the generator using the temporal datasets and the associated discriminators. The third method, TDGAN[35], uses the same reminding loss as CL-DSL, but has no model selection strategy, and instead of using metrics like Dist-FID mentioned earlier, it only keeps the last epoch's model.

Figure 5a shows the synthetic image quality comparisons among different methods after the final task. The synthetic samples are then used to train segmentation models. Figure 5b shows FID vs Dist-FID curves when DSL is learning task 4. Figure 5c compares the segmentation results from the different methods qualitatively. We find that Fine-Tuning produces more destructive synthetic images with PSNR 14.15 and SSIM 0.223, while CL-DSL generates superior quality images of each organ with PSNR 16.99 and SSIM 0.32. The segmentation model using high-quality synthetic images which are generated by CL-DSL obtains better segmentation results than other approaches. Table 5 compares the average performance of different methods for continual learning using the four tasks sequentially. It shows that CL-DSL achieves the best performance in terms of catastrophic forgetting in the continual learning. CL-DSL has a clear advantage over TDGAN[35] in most metrics. It is worth noting that the Fine-Tuning has worse image generation and segmentation performance compared to CL-DSL, even though Fine-Tuning learns all images from all temporal datasets. Fine-Tuning quickly forgets the features learned from the previous tasks during continual learning and can not continuously improve its performance since it does not use the reminding loss.

**Membership inference risk evaluation**

The ability to defend against malicious attacks is vital for building privacy-preserving machine-learning applications in medicine, especially in a distributed learning scenario. To validate the robustness of DSL under potential adversarial attacks, we extend to perform key ablation studies. We focus on the membership inference attack, which is a type of adversarial attack for machine learning models[36] that relates to privacy concerns.

In our study, we consider two types of attack settings to analyze the membership inference risk on the BraTS dataset. In setting 1, the attacker has access to a set of real images and the transformation-augmented synthetic database. Specifically, the attacker can access a set of real images from the OTHER center, including 300 images used for training the DSL (positive samples) and 300 images not used for training (negative samples). We evaluate the attacker's performance on a randomly selected set of real images from the CBICA and TCIA centers, consisting of 1000 positive and 1000 negative samples. We adopt a membership inference risk analysis similar to[37]. First, we calculate image similarity metrics between each real image and each synthetic image using the normalized root-mean-square error and perceptual distance. The perceptual distance is defined as 1-cosine of images' perceptual features, which are extracted from a pretrained ResNet50 model on the ImageNet database. These two metrics are normalized and summed to identify the closest synthetic image to each real image. Then, we use the two similarity metrics from the closest synthetic image as independent features to represent the real image and train SVM classifiers[38] on the 600 samples, and test on the 2000 samples. The linear support vector classifier (SVC) exhibits low testing accuracy, with an F1 score of 0.58, recall of 0.52, precision of 0.66, and AUC of 0.65. Similarly, the performance of the RBF-kernel SVC is also low, with an F1 score of 0.57, recall of 0.53, precision of 0.60, and AUC of 0.62. The membership inference attack faces two significant challenges. Firstly, 2D medical images from the same anatomical positions typically exhibit similar contextual information across different patients. This similarity can create numerous ambiguities during the attack. Secondly, the presence of affine transformations (such as rotation and scaling) and pixel-wise intensity differences in synthetic images make it more difficult to identify the correct match for the target. These challenges arise due to the absence of an ideal similarity measurement that can effectively address all these complexities.

In setting 2, the attacker has black-box access to the trained image generator through an API. By providing a real image's mask as input to the API, the attacker can obtain a corresponding synthetic image and compare it with the real image. We utilize two image similarity metrics (normalized root-mean-square error and perceptual distance) between paired real and synthetic images to perform a membership inference attack. Specifically, we use the same real data samples as in the previous setting to train and test SVM classifiers. By employing a linear support vector classifier (SVC), we obtain classification results on the testing samples, yielding an F1 score of 0.54, recall of 0.55, precision of 0.53, and AUC of 0.55. The utilization of an RBF-kernel SVC produces slightly lower results, with an F1 score of 0.45, recall of 0.38, precision of 0.54, and AUC of 0.55. The failed attack indicates that our trained generator has strong generalization and is not overfitting to the training data. Alternatively, a recent study proposed a membership inference attack technique[39] that worked on all major cloud-based machine learning services. The method first learned multiple "shadow models" that imitate the behavior of the target mode and then learned an attack model based on the outputs of the shadow models[39]. However, in our use case, the attacker can hardly learn good "shadow models" because of no access or knowledge of the discriminators and difficult training of GANs in practice[40].

It is therefore recognized that our DSL framework has a low membership inference risk. The privacy preservation in our framework could be further enhanced by incorporating additional security mechanisms like differential privacy[41] (all federated-learning methods in our experiments were trained without differential privacy). However, introducing differential privacy comes with a notable accuracy performance cost[42]. Introducing differential privacy to MM-DSL in the BraTS experiment makes the segmentation accuracy drop from Dice 0.829 ± 0.128 in Table 3 to 0.730 ± 0.201.

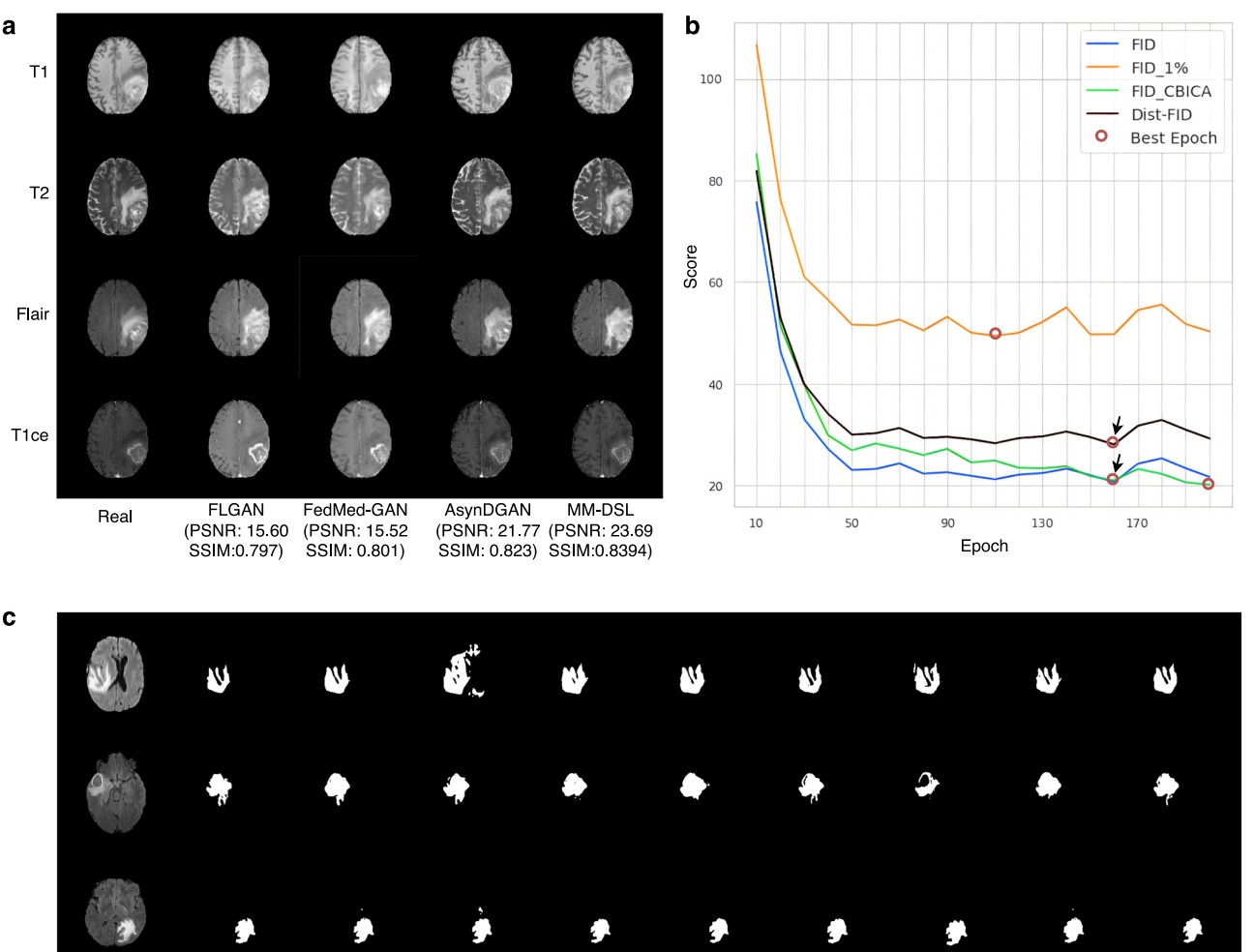

**Fig. 3 | Qualitative results of the multi-modality MRI brain experiment. a** One example of real multi-modality MRI brain images (Real), and the corresponding synthetic images generated by different methods (FLGAN, FedMed-GAN, AsynD-GAN, MM-DSL) with image quality metrics of peak signal-to-noise ratio (PSNR) and structural similarity index measure (SSIM). **b** The Dist-FID and FID score curves over the training epochs. The red circles indicate the best epoch for each method, while the arrows show the consistency of FID and Dist-FID scores. **c** Three examples of segmentation results for different methods vs the ground truth Label. The segmentation model learned from MM-DSL's synthetic data obtains more accurate results than other methods (Real-CBICA, Real-TCIA, Real-OTHER, FLGAN, FedMed-GAN, AsynDGAN) and is comparable to centralized learning (Real-All).

## Discussion

Learning from cross-silo private and heterogeneous data is a major challenge to enable large-scale, multi-center healthcare analytics. We have developed a GAN-based distributed architecture, termed Distributed Synthetic Learning (DSL), that achieves superior performance compared to the state of the art for heterogeneous and privacy-sensitive medical image data (see the tables in Results and the box plots in Supplementary Fig. 1). The superior performance of DSL is attributed to the distributed architecture containing one central generator and multiple distributed discriminators, as well as the novel efficient Dist-FID metric for selecting an optimal model. These two critical innovations facilitate DSL to characterize heterogeneous data properties, compensate for the misalignment of data modalities, and ensure privacy-preserving medical image analysis. In the following, we offer key insights into the advantage of DSL over other types of distributed architectures for building privacy-preserving distributed computational architectures for image generation.

### Table 4 | Quantitative evaluation of the missing-modality experiment

|  | Dice ↑ | HD95 (mm)↓ | SD (mm)↓ |
|---|---|---|---|
| Real-FedSeg | 0.839 ± 0.157 | 9.24 ± 12.42 | 1.75 ± 1.69 |
| Hetero-Real-FedSeg | 0.353 ± 0.144 | 65.84 ± 10.36 | 16.26 ± 3.66 |
| MM-DSL | 0.829 ± 0.128 | 11.50 ± 12.82 | 1.99 ± 1.86 |
| Hetero-MM-DSL | 0.795 ± 0.193 | 18.37 ± 19.44 | 4.00 ± 9.61 |

All methods learn from missing-modality brain MR data. The reported results (mean ± standard deviation) include Dice score, 95% quantile of Hausdorff distance (HD95), and average surface distance (SD). 'Real-FedSeg' indicates the segmentation model is trained by FedSeg from real complete-modality data in multiple data centers. 'MM-DSL' indicates the model is trained from the synthetic data generated by MM-DSL, which learns from complete-modality data. 'Hetero-' represents the adjusted methods for learning from missing-modality data. Hetero-MM-DSL leads to comparable segmentation results to the MM-DSL which learns from complete-modality data and is significantly better than the Hetero-Real-FedSeg.

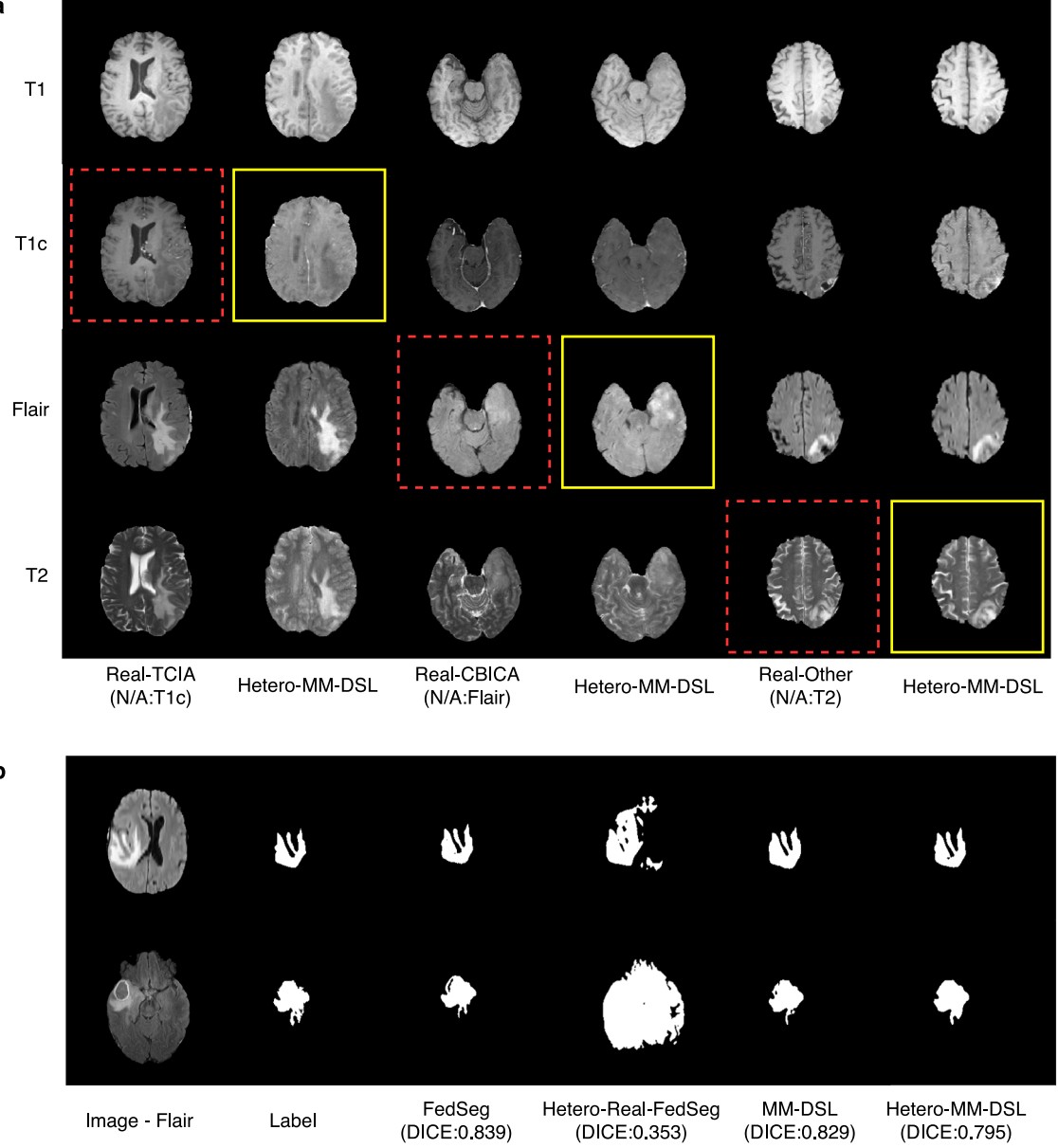

**Fig. 4 | Qualitative results of the missing-modality MRI brain experiment.** **a** Three examples of real brain images from three data centers respectively (Real-TCIA, Real-CBICA, Real-OTHER), and the corresponding synthetic images from our method (Heteo-MM-DSL). The red dash boxes indicate the missing modality and the yellow solid boxes indicate the completed synthetic image generated by DSL. **b** Two examples of segmentation results for different methods vs the ground truth Label. The segmentation model learned from Hetero-MM-DSL's synthetic data obtains more accurate results than Hetero-Real-FedSeg and does not have a clear performance drop compared to the complete-modality model (MM-DSL).

An essential feature that distinguishes DSL from other related approaches is the highly adaptable architecture capable of learning from heterogeneous data. The DSL framework is generalizable and can be used in various clinically important applications with significant data challenges introduced by varying imaging modalities, disease characteristics, and clinical acquisition protocols[43,44]. As seen in Fig. 1b, c, d, DSL works with multi-modality data, missing modalities, and temporal medical image datasets that often occur in clinical applications. DSL successfully addresses the data distribution misalignment in synthetic data generation applications. In the case of missing-modality data, DSL trained from misaligned modality data from different data centers can infer and complete the missing modalities and significantly outperforms the FedSeg by 55%. These DSL capabilities allow DSL to analyze large volumes of medical image data.

The image quality assessment with the Dist-FID metric represents an important contribution to this work. Although FID[32] has been widely used to evaluate image quality over an integrated dataset, measuring the overall image quality in multi-center data and privacy-sensitive scenarios remains largely unsolved. Simply applying the FID inception metric calculation in distributed private datasets is impractical due to data privacy restrictions. Previous distributed GAN methods[30,45] used public testing datasets to measure the synthetic image quality using the FID score. Such an approach inevitably leads to inaccuracies since the FID is unable to assess the image quality for each heterogeneous dataset, due to varying data sizes, modalities, and image quality. In our study, we find that the Dist-FID is an effective replacement of FID in distributed learning settings. From Figs. 2b, 3b, and 5b, we observe that the Dist-FID shows score consistency compared to FID for measuring image quality in each epoch. We can use Dist-FID to select the best

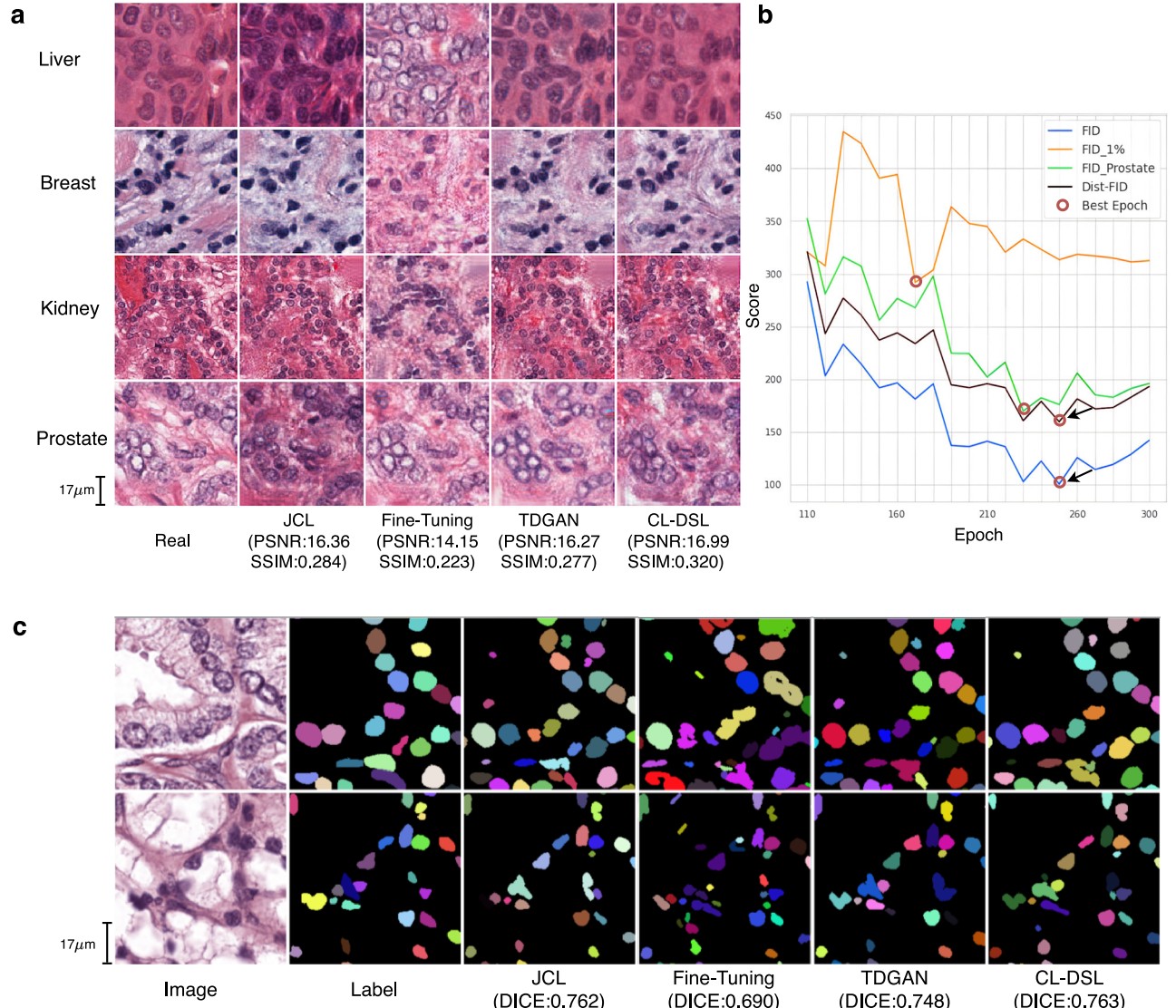

**Fig. 5 | Qualitative results of the continual learning experiment on the nuclei dataset. a** Four examples of real pathology images of four organs from different data centers (Real), and the corresponding synthetic images generated by different methods (JCL, Fine-Tuning, TDGAN, CL-DSL) with image quality metrics of peak signal-to-noise ratio (PSNR) and structural similarity index measure (SSIM). **b** The Dist-FID and FID score curves over the training epochs of the last task. The red circles indicate the best epoch for each method, while the arrows show the consistency of FID and Dist-FID scores. **c** Two examples of segmentation results for different methods after the final task vs the ground truth Label. The segmentation model learned from CL-DSL's synthetic data obtains more accurate results than the other methods (Fine-Tuning, TDGAN) and is comparable to centralized learning (JCL). The figures in each row of **a** are generated once for each method using the same mask input. The segmentation results are measured from distinct image patches that were regularly cropped in the preprocessing step. Repeating DSL by applying random transformations (e.g. scaling, shift, flip) on the input can generate more images with varied contexts and similar quality as shown in Supplementary Fig. 5.

**Table 5 | Quantitative results of nuclear segmentation**

| Method | Task1 (liver) | | Task2 (breast) | | Task3 (kidney) | | Task4 (prostate) | |
|---|---|---|---|---|---|---|---|---|
| | Dice ↑ | AJI ↑ | Dice ↑ | AJI ↑ | Dice ↑ | AJI ↑ | Dice ↑ | AJI ↑ |
| JCL | 0.6676 | 0.3420 | 0.7114 | 0.4457 | 0.7350 | 0.4814 | 0.7627 | 0.5184 |
| Fine-Tuning | 0.6676 | 0.3420 | 0.6950 | 0.4405 | 0.7142 | 0.4195 | 0.6902 | 0.4273 |
| TDGAN | 0.6676 | 0.3420 | 0.6961 | 0.4323 | 0.7164 | 0.4512 | 0.7481 | 0.4931 |
| CL-DSL | 0.6676 | 0.3420 | 0.7346 | 0.4638 | 0.7428 | 0.4605 | 0.7633 | 0.4828 |

All comparing methods learn GAN models in the continual learning setting and the segmentation models are trained from the corresponding synthetic data. The reported results include the mean of the Dice score and Aggregated Jaccard Index (AJI). Note that the results in the Task1 columns are the same among all the methods because the learning processes of Task1 are the same for different approaches. CL-DSL outperforms the other methods and obtains close performance to the centralized-learning method JCL.

model to extract the best quality generated images in each experiment. On the contrary, by using 1% of data from each data center or adopting all data from a single data center as a public testing dataset, the local FID calculated on these partial data does not result in the best quality synthetic images. By selecting the model with the best Dist-FID score we generate higher-quality synthetic data and the proposed DSL outperforms all other distributed GAN models in all three metrics (more score comparisons of FID and Dist-FID are in Supplementary Fig. 2). From the FID and Dist-FID curves in Figs. 2b, 3b, and 5b, the best model selected by Dist-FID is the same as the one selected by FID. When using the same model, the synthetic database and the downstream segmentation results are identical as well for Dist-FID and FID metrics.

Our study draws inspiration from FLGAN[45,46] and AsynDGAN[30] to address the privacy and multi-center data collection challenges. Due to the multiple technical innovations in DSL it consistently outperforms FLGAN, FedMed-GAN, and AsynDGAN in different scenarios. Since exchanging model parameters between the server and data centers (synchronizing the model parameters) is known to result in a severe performance drop when learning in heterogeneous environments[47,48], DSL does not synchronize model parameters as in standard FL-based approaches[45]. In contrast, our method directly computes the gradients and updates the model. In our study, DSL employs the AsynDGAN's topological structure but extends its adaptability and scalability for multiple modalities scenarios with the proposed Dist-FID module to select the optimal model. Furthermore, DSL uses an additional loss which makes DSL capable for continual learning. Taken together, DSL demonstrates its strong ability for multiple learning scenarios due to a reliable and robust synthetic data generator for downstream medical image analytics, while ensuring data privacy preservation from all data centers.

DSL ensures the security of private clinical data for the following two reasons. First, during model training, DSL transmits only synthetic images and the corresponding losses, which prevents the transmission of real DICOM images with sensitive patient information[49]. In the standard federated multi-node training where the same model is shared, an adversarial attack[50] can recover the original image data from model gradients. Such an attack does not work on DSL, because DSL does not have any model shared in any data center. The central server and all participants learn different models and no model parameters are exchanged. Second, during the synthetic data generation, random transformations (e.g., scaling, rotation, shift) can be applied to the input. This operation introduces additional image variances beyond the random dropout operations, providing an extra layer of data security protection. A key advance is that the randomly transformed synthetic dataset can lead to similar downstream segmentation performance while avoiding the generation of close-to-identical training samples. For example, in the brain tumor experiment, the segmentation metrics for training on the transformed synthetic database are Dice $0.838 \pm 0.121$, HD95 $14.68 \pm 15.74$, SD $2.26 \pm 1.93$, which are close to the MM-DSL results in Table 3. Note that all GAN-based methods being compared in Table 3 used not augmented synthetic data.

We also noticed that a previous study[51] showed the possibility of identification of anonymous 3D cranial MRI scans using face-recognition technology. However, our method was implemented using 2D networks and 2D images with privacy preservation. In our study, 2D medical images were indexed randomly and used independently without the original 3D information. Thus the generator does not learn any 3D information explicitly. In addition, the random transforms in the generation stage make 3D facial reconstruction impossible. Figures 2a and 3a show realistic-looking synthetic image examples and demonstrate the ability of our method to generate high-quality images. Meanwhile, Fig. 4a shows that in unconditioned areas the synthetic image can have very different semantic appearances (e.g., smaller brain ventricles) compared with the real image data. In summary, the generated synthetic database makes it impossible the 3D face reconstruction of a patient.

Our findings reflect several key insights on synthetic data augmentation that impact the performance of DSL. First, the architecture of DSL, inspired by image-to-image translation[52], provides randomness only in terms of several dropout layers in the generator network. This randomness does not offer significant variations to augment the synthetic data, thus we have applied separate data augmentation methods when generating a synthetic dataset for the downstream segmentation task. For instance, we applied random transformations to the input, such as scaling, shifting, flipping, and rotation, to generate more diverse images (see examples of synthetic images in Supplementary Figs. 3, 4, 5). Also, for datasets including multi-labels like the BraTS dataset, we can adopt a mixture of different brain skulls with tumor regions, introducing additional variations in the labels and synthetic datasets. This key operation differentiates the synthetic samples from the real samples while preserving the distribution of the real data. Second, the size of the original data affects the performance gain of the data augmentation method. When the size of all real data is small, scaling up the synthetic database by random transformations brings the benefit of increased variance in the synthetic database. We conducted an ablation study by using the histopathology data in a 4 data-center distributed learning setting similar to the cardiac and brain tasks. By generating a synthetic dataset twice the size of the real data, the augmented dataset used improves the segmentation performance (Dice: from 0.789 to 0.805, AJI: from 0.528 to 0.552). However, further increasing the size of the synthetic database does not benefit the downstream models. We also find that simply increasing the size of the synthetic database for the cardiac CTA and brain MRI tasks does not improve downstream tasks. A possible reason is that as the synthetic database becomes larger, repeating patterns in synthetic images may also accumulate and cause over-fitting in the training of downstream models. As a result, DSL applies the flexible data augmentation strategy to generate rich synthetic data for downstream segmentation tasks.

Our approach demonstrates potential generalization in handling multi-modality data with computational efficiency. In the multi-modality experiment, we constructed the MM-DSL with multiple discriminators at each center, where each discriminator focuses on a specific modality. While this design necessitates training using more parameters at each data center compared to using a single discriminator with multi-channel input for multi-modality data, it results in a superior generative model. Specifically, there is an improvement in dist-FID from 29.37 to 28.06, and downstream segmentation Dice from $0.821 \pm 0.16$ to $0.829 \pm 0.128$. This improvement is accompanied by a slight increase in overall training time (from 37.6 min per epoch to 40 min per epoch) and GPU memory usage for each data center (from 6.5 GB to 6.9 GB). Furthermore, our framework eliminates the requirement for training the generator at each data center, resulting in significant computational savings compared to other approaches such as FLGAN and FedMed-GAN. This design is generalizable and can handle the missing-modality scenario, so that it does not need to learn from a channel of empty modality data. The semantic correlations between different modalities are captured in the generator, which can synthesize multi-modality data as a multi-channel output. Thus, we did not introduce additional computations on the discriminator side to explicitly regularize the inter-modality connections. It is worth noting that our framework is not specifically designed to complete the missing modality in the real data domain[53]. The missing-modality completion in our study happens in the synthetic image domain and makes the complete-modality synthetic database statistically appropriate for downstream semantic segmentation tasks. As shown in Fig. 4a, the synthetic image does not match pixel-to-pixel the real image, since the input image does not provide constraints at every pixel.

The proposed framework can be extended to support many other downstream machine-learning tasks. Despite the DSL advances in handling heterogeneous data for clinical imaging applications, in this work, we limit our focus to multi-modality medical image generation and segmentation. We have not addressed the mixed use of labeled and unlabeled data to co-train the model for possible performance improvement[54,55] or domain generalization learning to adapt to unseen domains[56]. Also, combining imaging data with other forms of electronic health records such as clinical lab results or radiology reports into a united learning framework could be of substantial interest[57]. Additional controlling factors for the generator (instead of the segmentation masks), new techniques to generate both image and mask[58], and different downstream tasks can be further explored to assess the performance of DSL. For example, we can explore the use of bounding boxes or global labels to generate data for detection or classification, or even text-to-image generation[59]. DSL architecture is adaptive and can be used to provide insights for improved image analytics and understanding of disease from distributed heterogeneous medical data. This critical design can be helpful in learning large-scale medical foundation models[60–63].

## Methods

The study and results presented in this study comply with relevant ethical regulations and follow appropriate ethical standards in conducting research regarding the treatment of human subjects.

### Data collection and processing

We collected three categories of datasets described in Table 1 to evaluate our method: (1) multi-center cardiac computed tomography angiography (CTA); (2) multi-modality brain magnetic resonance imaging (MRI); (3) multi-organ histopathology images. The data heterogeneity lies in several aspects, including the number of samples, acquisition scanners, resolutions, geographic locations, modality (the missing-modality setting), and organs (the histopathology dataset). Supplementary Figs. 6, 7, and 8 show differences of some data samples among multiple centers.

For the Cardiac CTA data, we collected three public cardiac CTA datasets acquired from globally different institutes: the Multi-Modality Whole Heart Segmentation (MM-WHS) challenge dataset[64–66], Automated Segmentation of Coronary Arteries (ASOCA) challenge 2020 dataset[67,68], and MICCAI Coronary Artery Tracking Challenge 2008 (CAT08) dataset[69]. The heterogeneity of scanners and radiology protocols result in various range of voxel spacing and image quality. We only use the CTA data in the MM-WHS dataset, and denote this subset as WHS dataset in Table 1. The WHS data have manually annotated labels of seven whole heart substructures. We generated the annotations of the same substructures for CAT08 and ASOCA datasets by using a state-of-the-art whole heart segmentation algorithm[70] in the SenseCare research platform[71] and manually correcting gross errors. All the cardiac CTA data were resampled to isotropic 0.8 mm resolution. We used 200 and 1000 as the window level and width to transfer the Hounsfield units to intensity values in our experiments.

For the brain tumor MR images, we used 210 studies of glioblastoma (GBM) from the Brain Tumor Segmentation Challenge 2018 (BraTS18) training dataset[72–74]. The multi-modal MRI datasets were acquired with different clinical protocols and various scanners from 19 different institutions including the Center for Biomedical Image Computing and Analytics (CBICA), the Cancer Imaging Archive (TCIA), and other contributors (OTHER). We used 168 for training and validation and 42 for testing. Each case comprises four MRI modalities, including native (T1), T1 with gadolinium enhancing contrast (T1c), T2-weighted (T2), and T2 Fluid Attenuated Inversion Recovery (FLAIR). The ground truth annotation contains three types of tumor sub-regions including tumor core, enhancing tumor, and edema. All modalities have been aligned to a common space and resampled to 1mm isotropic resolution[74].

For the histopathology images, we used the multi-organ nuclei image dataset (Nuclei)[75]. Its public training set contains 30 digital microscopic tissue images from 30 patients and about 22,000 annotated nuclear boundaries in total (including both epithelial and stromal nuclei). These images of size $1000 \times 1000$ came from 18 different hospitals spanning seven organs. We selected four organs, the breast, kidney, liver, and prostate, to form a temporal dataset for evaluating continuous learning. Each dataset at a time point contains data from one of the organs. In our experiment, the training set of each center has 4 images from one organ. The testing set has 2 images per organ. In the preprocessing step[76], we first performed color normalization[77] for all images. Then, each image was divided into 16 ($4 \times 4$) overlapping tiles of size $286 \times 286$ to form the dataset in the experiment. Therefore, the training set has 64 images in each simulated data center and the testing set has 64 distinct image samples from different organs. In the training of the segmentation model, we used a tile size of $256 \times 256$, which is the same size as the input and output of the generator in DSL.

### Network architecture

Our proposed DSL is comprised of only one central generator and multiple distributed discriminators located in different local nodes. An overview of the proposed architecture is shown in Fig. 1. The central generator, denoted as $G$, takes task-specific inputs (e.g., segmentation masks in our use case) and generates synthetic images to fool the discriminators. Let $N$ denote the number of participating entities that collaborate in the learning framework, and $\mathbb{S}_j = \{(\mathbf{x}_i^j, \mathbf{y}_i^j)\}$ denote the local private dataset of size $|\mathbb{S}_j|$ at the $j$-th entity, where $\mathbf{x}$ is an auxiliary variable representing annotation, such as a class label or segmentation mask, $\mathbf{y}$ is the corresponding real image data, and $i \in \{1,...,|\mathbb{S}_j|\}$ is the sample index. The local discriminators, denoted as $D_j, j \in \{1, \ldots, N\}$, learn to differentiate between the local real images $\mathbf{y}_i^j$ and the synthetic images $\hat{\mathbf{y}}_i^j = G(\mathbf{x}_i^j)$ generated from $G$ based on $\mathbf{x}_i^j$. Our architecture ensures that $D_j$ deployed in the $j$-th medical entity only has access to its local dataset while not sharing any real image data outside the entity. Only synthetic images, annotations, and losses are transferred between the central generator and the distributed discriminators during the learning process.

**Central generator.** For segmentation tasks, the central generator is designed to generate images based on input masks so that the synthetic image and corresponding mask can be used as a pair to train a segmentation model. Here, an encoder-decoder ResNet[52], is adopted for $G$. It consists of nine residual blocks[78], two stride-2 convolutions for downsampling, and two transposed convolutions for upsampling. All non-residual convolutional layers are followed by batch normalization[79] and the ReLU activation. All convolutional layers use $3 \times 3$ kernels except the first and last layers that use $7 \times 7$ kernels.

**Distributed discriminators.** In our framework, each discriminator has the same structure as that in PatchGAN[52]. The discriminator classifies each of the overlapping patches of the input image as real or fake. Such architecture assumes patch-wise independence of pixels in a Markov random field fashion[52,80], and the patch is large enough ($70 \times 70$) to capture the difference in geometrical structures such as background and tumors.

The generator can learn the joint distribution of multiple isolated datasets through adversarial learning. Then, it can be used as an image provider to generate training samples for some downstream tasks. Assuming the distribution of synthetic images, $p_{\hat{\mathbf{y}}}$, is the same or similar to that of the real images, $p_{\text{data}}$, we can generate one large unified dataset, which approximately equals to the union of all the datasets in medical entities. In this way, all private image data from each entity are utilized without sharing. To evaluate the synthetic

images, we use the generated samples in segmentation tasks to illustrate the effectiveness of the proposed DSL.

## Objective function

The DSL is based on the conditional GAN[81]. The objective function is:

$$\min_G \max_{D_{1:N}} V(D_{1:N}, G)$$

$$= \sum_{j \in [N]} \boldsymbol{\pi}_j \Big\{ \mathbb{E}_{\mathbf{x} \sim s_j(\mathbf{x})} \Big[ \mathbb{E}_{\mathbf{y} \sim p_{\text{data}}(\mathbf{y}|\mathbf{x})} \log D_j(\mathbf{y}|\mathbf{x}) \tag{1}$$

$$+ \mathbb{E}_{\hat{\mathbf{y}} \sim p_{\mathbf{y}}(\hat{\mathbf{y}}|\mathbf{x})} \log(1 - D_j(\hat{\mathbf{y}}|\mathbf{x})) \Big] \Big\}$$

The goal of $D_j$ is to maximize Eq. (1), while $G$ minimizes it. In this way, the learned $G(\mathbf{x})$ with maximized $D(G(\mathbf{x}))$ can approximate the real data distribution $p_{\text{data}}(\mathbf{y}|\mathbf{x})$ and $D$ cannot tell 'fake' data from real. $\mathbf{x}$ follows a distribution $s(\mathbf{x})$. In this paper, We assume that the joint distribution $s(\mathbf{x}) = \sum_{j=1}^{N} \boldsymbol{\pi}_j s_j(\mathbf{x})$, where $s_j(\mathbf{x})$ is marginal distribution of $j$-th dataset and $\boldsymbol{\pi}_j$ represents the prior distribution. In the experiment, we set $s_j(\mathbf{x})$ to be a uniform distribution and $\boldsymbol{\pi}_j \propto |\mathbb{S}_j|$, resulting in a uniform distribution $s(\mathbf{x})$. For each sub-distribution, there is a corresponding discriminator $D_j$ which only receives data generated from prior $s_j(\mathbf{x})$. Similar to previous works[52,82], we incorporate noises by using Dropout[83] at several layers of the generator $G$ in both training and inference, instead of providing a Gaussian noise as input to the generator.

The losses of $D_j$ and $G$ are defined in Eq. (2) and Eq. (3), respectively.

$$L_{D_j} = \frac{1}{m} \sum_{i=1}^{m} \Big[ -\log D_j(\mathbf{y}_i^j|\mathbf{x}_i) - \log(1 - D_j(\hat{\mathbf{y}}_i^j|\mathbf{x}_i)) \Big], \tag{2}$$

$$L_G = \frac{1}{Nm \sum \boldsymbol{\pi}_j} \sum_{j=1}^{N} \boldsymbol{\pi}_j \sum_{i=1}^{m} [\log(1 - D_j(\hat{\mathbf{y}}_i^j|\mathbf{x}_i)) + \lambda_1 L_1(\mathbf{y}_i^j, \hat{\mathbf{y}}_i^j) + \lambda_2 L_P(\mathbf{y}_i^j, \hat{\mathbf{y}}_i^j)]. \tag{3}$$

where $m$ is the minibatch size. The $L_G$ contains perceptual loss ($L_P$)[84] and $L_1$ loss besides the adversarial loss. In this study, $G$ and $D_j$ are not on the same server and thus Eq. (3) needs to be split into two parts Eq. (4) and Eq. (5) in order to back-propagate the losses to $G$.

$$L_{G_j} = \frac{1}{m} \sum_{i=1}^{m} [\log(1 - D_j(\hat{\mathbf{y}}_i^j|\mathbf{x}_i)) + \lambda_1 L_1(\mathbf{y}_i^j, \hat{\mathbf{y}}_i^j) + \lambda_2 L_P(\mathbf{y}_i^j, \hat{\mathbf{y}}_i^j)]. \tag{4}$$

$$\nabla_{\hat{\mathbf{y}}} = \frac{1}{N \sum \boldsymbol{\pi}_j} \sum_{j=1}^{N} \boldsymbol{\pi}_j [\nabla_{\hat{\mathbf{y}}^j}], \tag{5}$$

where $\nabla_{\hat{\mathbf{y}}^j} = \partial L_{G_j}/\partial \hat{\mathbf{y}}^j$ is computed at node $D_j$ based on loss in Eq. (4) and then sent back to $G$ for aggregation (Eq. (5)). The learning process is summarized in Supplementary Algorithm 1. We trained 200 epochs for all tasks and updated each discriminator once in each training iteration. The gradient-based updates can adopt different gradient-based learning rules. We used Adam optimizer[85] with a learning rate of 0.0002 in our experiments.

## Extension for multi-modality datasets

For a use case of multi-modality data, assuming $c$ modalities, the local data center $j$ has a set of multi-modality image $y_i^j = (\mathbf{y}_{i,1}^j, \ldots, \mathbf{y}_{i,c}^j)$ associated with each label image $\mathbf{x}_i^j$. A simple way of handling the multi-modality image in our framework would be treating the $c$ modalities of one sample as a $c$-channel image. Thus the only change needed is the number of channels of the input layer of $D$ and an output layer of $G$. In this setting, the learning task of $D$ could be easier and converge very fast since different modalities have different contrast patterns, and

more information can be used to differentiate the real and the 'fake' data. However, the task of $G$ may become more challenging to learn. It is because, on one hand, the $G$ needs to learn more complex data distribution to generate multiple modalities with different contrasts. On the other hand, the easily-learned $D$ may learn some trivial discriminative features and thus cannot provide helpful feedback to $G$ to guide its learning.

To balance the task difficulty of the $G$ and $D$'s, we extend our framework by deploying multiple discriminators at each entity. Every single modality has its discriminator in one data center, and the $G$ receives losses from the multiple $D$s for a multi-modality data sample. In this way, each $D$ can focus on learning discriminative features for one specific modality and provide more meaningful feedback to $G$. The objective function can be extended from Eq. (1) as:

$$\min_G \max_{D_{1:N}^{1:c}} V(D_{1:N}^{1:c}, G)$$

$$= \sum_{j \in [N]} \boldsymbol{\pi}_j \Big\{ \mathbb{E}_{\mathbf{x} \sim s_j(\mathbf{x})} \sum_{k=1}^{c} \Big[ \mathbb{E}_{\mathbf{y}_k \sim p_{\text{data}}(\mathbf{y}_k|\mathbf{x})} \log D_{j,k}(\mathbf{y}_k|\mathbf{x}) \tag{6}$$

$$+ \mathbb{E}_{\hat{\mathbf{y}}_k \sim p_{\mathbf{y}}(\hat{\mathbf{y}}_k|\mathbf{x})} \log(1 - D_{j,k}(\hat{\mathbf{y}}_k|\mathbf{x})) \Big] \Big\},$$

where $D_{j,k}$ represents the discriminator for the $k$-th modality at the center $j$.

Besides, another advantage of the proposed multi-modality framework is that it enables learning from missing modality data. Let $C_j$ denote the set of index of available modality for center $j$, if data center $j$ misses the $k$-th modality for example, then $C_j = \{1, \ldots, k-1, k+1, \ldots, c\}$. In this case, center $j$ only needs to deploy $c-1$ discriminators during the learning. The learning process has no difference except that it only collects losses of available discriminators for $C_j$ to update the $G$ and only use a subset of the synthetic images $\{\hat{\mathbf{y}}_k^j|k \in C_j\}$ to update the corresponding $\{D_{j,k}|k \in C_j\}$ in center $j$. Because the discriminators for different modalities in different entities are all independent, the $G$ can still learn to generate all modalities, assuming that the missing modality in one center is available in some other data centers. The loss function of $D$ is the same, while the loss function of $G$ can be adjusted as the following:

$$L_G = \frac{1}{Nm \sum \boldsymbol{\pi}_j} \sum_{j=1}^{N} \boldsymbol{\pi}_j \sum_{i=1}^{m} \sum_{k \in C_j} \Big[ \log(1 - D_{j,k}(\hat{\mathbf{y}}_{i,k}^j|\mathbf{x}_i)) \tag{7}$$

$$+ \lambda_1 L_1(\mathbf{y}_{i,k}^j, \hat{\mathbf{y}}_{i,k}^j) + \lambda_2 L_P(\mathbf{y}_{i,k}^j, \hat{\mathbf{y}}_{i,k}^j) \Big].$$

After training, the learned $G$ can act as a synthetic image provider to generate multi-modality images from the conditional variable, a mask image. As a result, it can also be used for missing modality completion. For instance, if a data center has data $(\mathbf{y}_1, \ldots, \mathbf{y}_{k-1}, \mathbf{y}_{k+1}, \mathbf{y}_c)$ with the $k$-th modality missing and the corresponding mask image $\mathbf{x}$, we can use the synthetic image at the $k$-th channel of $G(\mathbf{x})$ as a substitute. Our approach is different from the existing methods that predict the target modality from another modality[53,86] in the sense that it can generate multiple modalities to handle randomly missing modality problems, and thus does not require a specific model for specific modality pair for the input and output.

## Extension for temporal datasets

Another variation of DSL contains a central generator and multiple distributed temporary discriminators located in data centers. Suppose the training starts at time $t-1$ with $K_{t-1}$ online local data centers. The central generator $G_{t-1}$ learns the distribution of all online inputs and outputs synthetic images. The local discriminators, $\{D_{t-1}^1, \ldots, D_{t-1}^{K_{t-1}}\}$ learn to identify the synthetic images from the local real images. At time $t$, the new data centers are online and the real data and discriminators of $t-1$ are no longer available. The central generator $G_t$

tries to learn the distribution of new data and retain the mixture distribution learnt from previous data. The learning of new data is achieved by a digesting loss and the memory of previously learnt knowledge is kept by using a reminding loss.

We assume the conditional distribution is consistent over time. The loss function of TDGAN consists of two parts:

$$V_t(G_t, D_t^{1:K_t}) = \min_{G_t} L_{\text{Digesting}} + \lambda \cdot L_{\text{Reminding}}$$

$$\text{Digesting Loss}: L_{\text{Digesting}} \triangleq \max_{D_t^{1:K_t}} \sum_{j=1}^{K_t} \pi_t^j \mathbb{E}_{\mathbf{x} \sim s_t^j(\mathbf{x})} \Big\{ \mathbb{E}_{\mathbf{y} \sim p_{\text{data}}(\mathbf{y}|\mathbf{x})} [\log D_t^j(\mathbf{y}|\mathbf{x})]$$
$$+ \mathbb{E}_{\hat{\mathbf{y}}_j \sim p_y(\hat{\mathbf{y}}_t^j|\mathbf{x})} [\log(1 - D_t^j(G_t(\mathbf{x})|\mathbf{x}))] \Big\}$$

$$\text{Reminding Loss}: L_{\text{Reminding}} \triangleq \mathbb{E}_{\mathbf{x} \sim s_{t-1}(\mathbf{x})} \mathbb{E}_{\hat{\mathbf{y}} \sim p_y(\hat{\mathbf{y}}|\mathbf{x})} [\| G_t(\mathbf{x}) - G_{t-1}(\mathbf{x}) \|^2]$$

$$\text{(8)}$$

The digesting loss, $L_{\text{Digesting}}$, utilizes the mixture cross-entropy loss term to supervise the generator to learn from the new data at time $t$. The reminding loss, $L_{\text{Reminding}}$, is formulated as a squared norm loss to enforce the generator to memorize the learned distribution of past data.

## Distributed FID for image quality measurement

The Frechet Inception Distance (FID)[32] has been widely used to evaluate the image qualify by calculating the distance between the statistics of feature vectors of the real and generated images. The definition of FID is:

$$\text{FID} = ||\boldsymbol{\mu}_1 - \boldsymbol{\mu}_2||^2 + \text{Tr}(\boldsymbol{\sigma}_1 + \boldsymbol{\sigma}_2 - 2^*\sqrt{\boldsymbol{\sigma}_1 {}^* \boldsymbol{\sigma}_2}), \tag{9}$$

where $\boldsymbol{\mu}_1$ and $\boldsymbol{\mu}_2$ refer to the feature-wise mean of the real and generated images, $\boldsymbol{\sigma}_1$ and $\boldsymbol{\sigma}_2$ are the covariance matrices for the real and generated feature vectors, Tr refers to the trace operation in linear algebra.

Though FID is an ideal metric to find the best model when training a GAN[26,87], we are unable to compute one FID score in distributed learning because a joint set of the isolated real data does not exist. Therefore, we propose a new metric named distributed FID (DistFID) to calculate the weighted average distance between each real dataset and the synthetic database. The DistFID is defined as:

$$\text{DistFID} = \sum_j^N \mathbf{w}_j (||\boldsymbol{\mu}_1^j - \boldsymbol{\mu}_2||^2 + \text{Tr}(\boldsymbol{\sigma}_1^j + \boldsymbol{\sigma}_2 - 2 * \sqrt{\boldsymbol{\sigma}_1^j * \boldsymbol{\sigma}_2})) \tag{10}$$

in which each of the $N$ entities host a dataset $\mathbb{S}_j$ of size $|\mathbb{S}_j|$ with feature statistics $(\boldsymbol{\mu}_1^j, \boldsymbol{\sigma}_1^j)$. The weight of each center $\mathbf{w}_j = |\mathbb{S}_j| / \sum_{j=1}^N |\mathbb{S}_j|$. At the beginning of training DSL, each client center sends the feature-wise statistics $(\boldsymbol{\mu}_1^j$ and $\boldsymbol{\sigma}_1^j)$ to the central center. Then, the central center can use the synthetic images and compute the DistFID value based on Eq. (10) to evaluate the generator. We validated the consistency between the FID and DistFID scores in Supplementary Fig. 2.

## Learning of downstream task

In this study, we used segmentation as the downstream machine-learning task and also evaluated a classification task. After obtaining a well-learned image generator from the DSL, we can generate synthetic medical images from mask (label) images. In our experiments, to fairly compare the effect of the synthetic images with the real samples, we adopted the same U-Net[88] as the segmentation model and VGG[34] network as the classification model to learn on different sets of 2D images. During training the downstream task, we withheld 20% samples from the training data as the validation set to select the model with the best Dice score to test. We used Adam optimizer with a learning rate of 0.01 to learn segmentation in our experiments. For the cardiac CTA and brain MRI segmentation tasks, a combination of cross-entropy (CE)

and Dice was used as the loss function. For the nuclear segmentation task, CE loss was used. For the classification task, the binary cross-entropy loss was used. We inferred every 2D image in testing, and for the cardiac CTA and brain MRI data we computed the 3D metrics by stacking up the 2D images for the same subject, which are reported in Results. Note that, the reported standard deviations in the Results section were computed with the degrees of freedom equaling the number of samples.

## Quantitative metrics

The Dice score (Dice) and 95% quantile of Hausdorff distance (HD95) are adopted to evaluate the segmentation performance on cardiac CTA and BraTS18[74]. The Dice score measures the overlap between ground-truth mask $\mathcal{G}$ and segmented result $\mathcal{S}$. It is defined as

$$\text{Dice}(\mathcal{G}, \mathcal{S}) = \frac{2|\mathcal{G} \cap \mathcal{S}|}{|\mathcal{G}| + |\mathcal{S}|} \tag{11}$$

The Hausdorff Distance (HD) evaluates the distance between boundaries of ground-truth and segmented masks:

$$\text{HD}(\mathcal{G}, \mathcal{S}) = \max\{\sup_{\mathbf{u} \in \partial\mathcal{G}} d(\mathbf{u}, \partial\mathcal{S}), \sup_{\mathbf{v} \in \partial\mathcal{S}} d(\mathbf{v}, \partial\mathcal{G})\} \tag{12}$$

where $\partial$ means the boundary operation, $d(\mathbf{u}, \partial\mathcal{S}) = \inf_{\mathbf{v} \in \partial\mathcal{S}} \| \mathbf{u} - \mathbf{v} \|_2$ is minimum distance from vertex $\mathbf{u}$ to surface $\partial\mathcal{S}$ and sup represents the supremum and inf the infimum. Because the Hausdorff distance is sensitive to outliers in $\mathcal{G}$ or $\mathcal{S}$, we use the 95% quantile Hausdorff distance (HD95):

$$\text{HD95}(\mathcal{G}, \mathcal{S}) = \max\{\sup_{\mathbf{u} \in \partial\mathcal{G}}^{95} d(\mathbf{u}, \partial\mathcal{S}), \sup_{\mathbf{v} \in \partial\mathcal{S}}^{95} d(\mathbf{v}, \partial\mathcal{G})\}, \tag{13}$$

where the sup[95] is the 95%-th maximum value. In addition, we report the average Surface Distance (SD) as follows:

$$\text{SD}(\mathcal{G}, \mathcal{S}) = \frac{1}{2} \left\{ \frac{1}{|\partial\mathcal{G}|} \sum_{\mathbf{u} \in \partial\mathcal{G}} d(\mathbf{u}, \partial\mathcal{S}) + \frac{1}{|\partial\mathcal{S}|} \sum_{\mathbf{v} \in \partial\mathcal{S}} d(\mathbf{v}, \partial\mathcal{G}) \right\}. \tag{14}$$

For nuclei segmentation, we utilize the object-level Dice[89] and the Aggregated Jaccard Index (AJI)[75]:

$$\text{AJI}(\mathcal{G}, \mathcal{S}) = \frac{\sum_{i=1}^{n_\mathcal{G}} |\mathcal{G}_i \cap \mathcal{S}(\mathcal{G}_i)|}{\sum_{i=1}^{n_\mathcal{G}} |\mathcal{G}_i \cup \mathcal{S}(\mathcal{G}_i)| + \sum_{k \in \mathcal{O}} |\mathcal{S}_k|} \tag{15}$$

where $n_\mathcal{G}$ is the number of ground-truth objects in $\mathcal{G}$, $\mathcal{S}(\mathcal{G}_i)$ represents the segmented object that has maximum overlap with $\mathcal{G}_i$ with regard to the Jaccard index, and $\mathcal{O}$ is the set containing segmentation objects that have not been assigned to any ground-truth object.

## Reporting summary

Further information on research design is available in the Nature Portfolio Reporting Summary linked to this article.

## Data availability

The datasets used in this study are all publicly available. The WHS data are available through the MM-WHS Challenge[64–66] at https://zmiclab.github.io/zxh/0/mmwhs/. The ASOCA dataset[67,68] is available at https://asoca.grand-challenge.org/. The CAT08 dataset can be obtained by contacting the challenge organizers[69]. The WHS masks for the ASOCA and CAT08 can be obtained at https://github.com/tommy-qichang/DSL_All_Code/tree/main/data. The brain data are available through BraTS 2018 Challenge[72–74] at https://www.med.upenn.edu/sbia/brats2018.html. The nuclei dataset[75] is available as the MoNuSeg Challenge training set at https://monuseg.grand-challenge.org/Data/.

Source data are provided with this paper. Specifically, we provide the raw data of Fig. 2b, Fig. 3b, Fig. 5b, Table 2, Table 3, Table 4, and Table 5 to reproduce the plots and statistics of results of this study in a public repository[90] at https://github.com/tommy-qichang/DSL_All_Code/tree/main/data. In the file SourceData.xlsx, each sheet represents the source data of a figure or a table.

## Code availability

DSL is implemented in Python 3.7 using PyTorch framework 1.6.0[91]. It is implemented in a stand-alone environment based on the PyTorch implementation of pix2pix[52] (https://github.com/junyanz/pytorch-CycleGAN-and-pix2pix), and a distributed environment based on FedML[92]. The implementation of comparing method FedSeg can be found at https://fedsegment.github.io/home. The AsynDGAN[30] can be found at https://github.com/tommy-qichang/AsynDGAN. The FLGAN[45,46] and FedMed-GAN[23] are re-implemented in the FedML framework. The source codes of DSL, FLGAN, FedMed-GAN, and the segmentation used in this study can be found in a public repository[90] at https://github.com/tommy-qichang/DSL_All_Code.

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

## Acknowledgements

We thank Qing Xia and Wenji Wang for helping to generate segmentation masks for CAT08 and ASOCA datasets using their method[70]. This work was partially supported by grants from National Science Foundation (1747778, 1849238, 1951890, 2212301, 2235405) (D.N.M.), and the Centre for Perceptual and Interactive Intelligence (CPII) Ltd under the Innovation and Technology Commission (ITC)'s InnoHK (H.L. and S.Z.). H.L. and S.Z. are PI and co-PI of the CPII.

## Author contributions

Q.C., Z.Y., and M.Z. jointly designed experiments, interpreted the data, and wrote the paper. Q.C. and Z.Y., implemented experiments with the help of H.Q. and X.H. H.Z., L.B., and S.A. helped to review and analyze the experimental results. H.L., S.Z., and D.N.M. supervised the project. All authors discussed the results and implications and commented on the manuscript at all stages.

## Competing interests

The authors declare no competing interests.
