## [Peer Review File · Nature Communications]

nature portfolio

Peer Review FileReviewer comments, first round

Reviewer #1 (Remarks to the Author):

The proposed work conducted a distributed synthetic learning problem in the healthcare system, and proposed a distributed generative adversarial network to address such problem. In detail, three main methods, standard distributed discriminators with a central generator, multi-discriminators for multi-modality data in one participant, and a local loss function revision for continuous learning, are investigated and proposed to solve the privacy preservation and data heterogeneity in DSL. To verify the effectiveness of the proposed method, several real life experiments are conducted, and obvious performance gains are provided to some benchmarks. However, there are several major concerns from the reviewer, listed below.

1. The technique contribution is not that novel. Using DSL in the healthcare system is always a heated topic, however, the listed references are not the newest. In addition, the compared benchmarks are not very new. To verify the effectiveness of the proposed algorithm, the reviewer suggest the authors can provide additional comparisons with other state-of-the-arts if possible.
2. Following comment 1, using distributed FID seems to be a major different for other distributed GANs. Although the authors have provided several comparisons between FID and DistFID, the reviewer would like to check more results on the Dice, HD95 and SD using FID.
3. Another major concern is related to "privacy preservation". Even though real data is not transmitted among different utilities, some attributes of the real data can still be discovered from the transmitted gradients or losses. And, some privacy attacking method, such as inference membership attack, can also lead to privacy leakage by consulting the generated data samples. The reviewer suggest the authors can enhance the related content, and provide or discuss some privacy-enhancing methods, such as differential privacy, to address such concern.
4. Some design details are missing. For example, how to use DistFID in the proposed method should be introduced clearly. Why use dropout to replace random Gaussian noises as input?
5. In terms of multi-modal dataset, building multiple discriminators will increase a large or overtaken computation overhead. Is there any way to alleviate such burden? In addition, some connections between different discriminators of one participant should be build up as they all from one dataset.

Overall, the proposed work has some contributions with noteworthy results, but the authors may revise some parts of the manuscript, and the reviewer would like to check the changes if they are well addressed.

Reviewer #2 (Remarks to the Author):

This study proposed a distributed synthetic learning (DSL) architecture to address patient privacy and heterogeneity concerns in multi-center data scenarios. GAN-based synthetic learning is used as a main technique to shield the center information. The architecture contains one central generator and multiple node discriminators. The evaluation is conducted on three segmentation applications.

- (1) Since DSL is designed to address medical privacy and heterogeneity challenges from multi-center datasets, it should be explained clearer how the heterogeneity affects the data and the model performance. At least, a sample of difference among multiple centers should be displayed.
- (2) Only experiments for segmentation are reported, to fully evaluate the ability of DSL, it would be better to report experiments on some other types of applications, e.g., image reconstruction or classification.
- (3) It seems that all the models (e.g., synthesis and segmentation models) are implemented in 2D convolutional networks, but some of the data (e.g., BraTS18) are 3D images. It would be better to conduct experiments on 3D networks for these data.
- (4) In Figure 3, it is not professional to display brain images in that direction, please display them like that in Figure 1C.

datasets (single modality), the authors state that DSL employs AsynDGAN's structure (Line 241), but DSL and AsynDGAN show different performances in Table 1. This is confusing and needs clarification. Also, subsection "Extension for temporal datasets" is very confusing because the authors mainly introduce the optimization objective of a baseline model, instead of the proposed model. Their relationship should be clearly stated.

4 The authors report comparison results between CL-DSL and FLGAN (Line 179) but it is hard to figure out where these numbers and comparisons come from. According to the stated experimental setting in Lines 169-175, it seems that such comparisons were not planned/described. This makes the results in this paragraph (especially Lines 189-191) hard to follow.

5 This reviewer cannot access the source code for evaluation because the provided link (<https://rutgers.app.box.com/folder/147492331099>) is private.

6 It is unclear what are the key takeaways from Figures 2C, 3C, 4B, and 5C.

7 This reviewer cannot understand the undefined term $G_t(u,y)$ in Eq 8. Why a generator can be fed with a real image y and some noise? The whole Eq 8 was copied from the authors' previous publication but there exist definition inconsistencies, making it hard to follow.

This reviewer also has several specific concerns.

1 The authors select FedSeg and Hetero-FedSeg as two major baseline methods for Hetero-MM-DSL, but neither references nor descriptions were provided for review. The authors also need to provide justifications on why FedSeg was selected as a baseline. It is also unclear whether Hetero-FedSeg uses synthetic images to learn the model or not. If not, it would be better to name it something like Hetero-Real-FedSeg.

2 In the caption of Table 5, the authors mentioned "Real- indicates the model ..."; however, there is no model with the prefix "Real-" in Table 5.

3 Redundancy needs to be removed for subsection "Data collection and processing". Many details have been introduced in the Data subsection. The authors can probably merge these two subsections.

4 The authors claim (Line 291) that the BraTS18 dataset was split to 170 and 40 for training and testing; however, these seem inconsistent with the numbers in Table 1 (168 vs 42).

5 The authors use "MM-DSL" to refer to the DSL architecture that considers the setting of multi-modality images (Line 127); however, the authors never use this name in Results and Methods, but used "Hetero-MM-DSL" for the missing modality architecture. It seems that "DSL" in Tables 3 and 4 should be "MM-DSL", and all "DSL" refer to the full modality architecture should be renamed accordingly.

6 Cell "40 (6360)ss" in Table 1 needs to be corrected.

7 Function $G_s(x)$ not defined before use (Line 405).

8 " t_4 " in Line 168 seems unnecessary and can be removed.

9 There are typos. For example, Line 230 "higher qualify" -- > "higher quality"

Response to reviewers' comments for NCOMMS-22-35277

We appreciate the reviewers' comments on our manuscript. We have made all requested responses to the comments raised by the reviewers. All modifications we made are highlighted in blue in the revised manuscript. In the following, we provide detailed answers to each Reviewer's comments. All table indices, section indices, reference indexes, and page numbers refer to the revised manuscript of NCOMMS-22-35277A.

Reviewer #1 (Remarks to the Author):

The proposed work conducted a distributed synthetic learning problem in the healthcare system, and proposed a distributed generative adversarial network to address such problem. In detail, three main methods, standard distributed discriminators with a central generator, multi-discriminators for multi-modality data in one participant, and a local loss function revision for continuous learning, are investigated and proposed to solve the privacy preservation and data heterogeneity in DSL. To verify the effectiveness of the proposed method, several real life experiments are conducted, and obvious performance gains are provided to some benchmarks. However, there are several major concerns from the reviewer, listed below.

1. The technique contribution is not that novel. Using DSL in the healthcare system is always a heated topic, however, the listed references are not the newest.

In addition, the compared benchmarks are not very new. To verify the effectiveness of the proposed algorithm, the reviewer suggests the authors can provide additional comparisons with other state-of-the-art if possible.

[Response] Thank you for your comments. We have added more recent related works to the reference list, including [20],[25],[51-54]. As seen below, we have also added comparison results to the FedMed-GAN method [20] in Table 2 and Table 3 of the Results section. Finally, we have discussed more related references [40-48,55] as other reviewers requested as well.

[Response] Thank you for this comment. We have modified the manuscript and as we mention in Discussion section, Figs. 2(B), 3(B) and 5(B) show that the trend of Dist-FID is consistent with the FID as the metric of synthetic image quality. Since it is not feasible to compute the real FID score in practice, which requires access to all images in all clients, we compute Dist-FID instead and select the best model of the generator according to the Dist-FID score. In our experiments, the best model is the same as the one using the FID score. We thus have added sentences in the Discussion section for clarification. "From the FID and Dist-FID curves in Figs. 2(B), 3(B), and 5(B), the best model selected by Dist-FID is the same as the one selected by FID.. When using the same model, the synthetic database and the downstream segmentation results are identical as well for Dist-FID and FID metrics."

3. Another major concern is related to "privacy preservation". Even though real data is not transmitted among different utilities, some attributes of the real data can still be discovered from the transmitted gradients or losses.

And, some privacy attacking method, such as inference membership attack, can also lead to privacy leakage by consulting the generated data samples. The reviewer suggests the authors can enhance the related content, and provide or discuss some privacy-enhancing methods, such as differential privacy, to address such concern.

[Response] Thank you for the insightful comments. We have added ablation studies and additional discussion on privacy and security in the revision.

In the Results section, we added the following paragraphs:

“Membership Inference Attack Evaluation The ability to defend against malicious attacks is vital for building privacy-preserving machine-learning applications in medicine, especially in a distributed learning scenario. To validate the robustness of DSL under potential adversarial attacks, we extend to perform key DSL ablation studies. We focus on the membership inference attack which is a type of adversarial attack for machine learning models [35] that relates to privacy concerns. In our study, given a real data record (images), and black-box access to the trained image generator (setting 1) or the synthetic database (setting 2), an attacker would like to determine if the real record was in the model's training dataset. We analyzed two important settings for the membership adversarial attack using the BraTS dataset.

In the first attack setting, only the trained generator is accessible as an API to attackers. An attacker can obtain the synthetic image given an input mask and compare it with the corresponding real image. The hypothesis is that a trained sample is more similar to its corresponding synthetic image than an unseen sample, and thus one attacker can utilize the similarity between the paired real and synthetic images to achieve a membership inference attack. We adopted two image similarity metrics, including the cosine distance score of the perceptual features from pretrained resnet50 model on ImageNet (perceptual similarity = 1 - cosine distance) and the structural similarity index measure (SSIM). The statistics of perceptual similarity for 11,349 trained samples vs 2,730 unseen samples are 0.996962 ± 0.000195 vs 0.996963 ± 0.000194 ($p=0.8946$). The statistics of SSIM for the trained samples vs unseen samples are 0.803466 ± 0.049409 vs 0.804076 ± 0.048799 ($p=0.5615$). From these details, there

is no statistical difference between trained samples and unseen samples in terms of these similarity metrics. Alternatively, a recent study proposed an effective membership inference attack technique [36] that worked on all major cloud-based machine learning services. The method first learned multiple “shadow models” that imitate the behavior of the target mode and then learned an attack model based on the outputs of the shadow models [36]. However, in our use case, the attacker can hardly learn good “shadow models” because of no access or knowledge of the discriminators and difficult training of GANs in practice [37]. It is therefore recognized that an effective membership attack is not feasible for our DSL framework.

In the second attack setting, only the transformed synthetic database is accessible to attackers. An attacker can use image retrieval techniques to search for the most similar match for a given real image, and then infer the membership based on the image similarity. In this second setting, we used a fast (approximated) nearest neighbor method <https://github.com/pixelogik/NearPy> and the perceptual features from pretrained resnet50 to implement the image retrieval. We assume that the attacker could access the other site (975 training samples) in the brain task and another 975 unseen samples to train an attack model (classifier) based on two features, including the perceptual similarity and SSIM between the real image and the corresponding retrieved synthetic image. The attack model was then tested on randomly selected 1,750 training samples from the TCIA and CBICA sites and 1,750 unseen samples. The classification results have accuracy 0.5 and F1 score 0.5 using a linear support vector classification (SVC), accuracy 0.47 and F1 score 0.52 by an RBF-kernel SVC [38]. We see that the synthetic database with random transformations demonstrates its robustness to defend the membership inference attacks. The privacy preservation in our framework could be further enhanced by incorporating additional security mechanisms like differential privacy [39] (all federated-learning methods in our experiments were trained without differential privacy). However, introducing differential privacy comes with a notable accuracy performance cost [40]. Introducing differential privacy to MM-DSL in the BraTS experiment makes the segmentation accuracy drop from Dice 0.829 ± 0.128 in Table 3 to 0.730 ± 0.201 .”

We also added the following paragraphs in the Discussion section:

“DSL ensures the security of private clinical data for the following two reasons. First, during model training, DSL transmits only synthetic images and the corresponding losses which prevents the transmission of real DICOM images with sensitive patient information [47]. In the standard federated multi-node training where the same model is shared, an adversarial attack [48] can recover the original image data from model gradients. Such an attack does not work on DSL, because DSL does not have any model shared in any data-center. The central server and all participants learn different models and no model parameters are exchanged.

Second, during the synthetic data generation, random transformations (e.g., scaling, rotation, shift) can be applied to the input. This operation introduces additional image variances beyond the random dropout operations, providing an extra layer of data security protection. A key advance is that the randomly transformed synthetic dataset can lead to similar downstream segmentation performance, while avoiding generation of close-to-identical training samples. For example, in the brain tumor experiment, the segmentation metrics for training on the

transformed synthetic database are Dice 0.838 ± 0.121 , HD95 14.68 ± 15.74 , SD 2.26 ± 1.93 , which are close to the MM-DSL results in Table 3. Note that all GAN-based methods being compared in Table 3 used not augmented synthetic data.

We also noticed that a previous study [49] showed the possibility of identification of anonymous 3D cranial MRI scans using face-recognition technology. However, our method was implemented using 2D networks and 2D images with privacy preservation. In our study, 2D medical images were indexed randomly and used independently without the original 3D information. Thus the generator does not learn any 3D information explicitly. Additionally, the random transforms in the generation stage makes 3D facial reconstruction impossible. Fig. 2(A) and Fig. 3(A) show realistic-looking synthetic image examples and demonstrate the ability of our method to generate high-quality images. Meanwhile, Fig. 4(A) shows that in unconditioned areas the synthetic image can have very different semantic appearances (e.g., smaller brain ventricles) compared with the real image data. In summary, the generated synthetic database does makes it impossible the 3D face reconstruction of a patient.”

4. Some design details are missing. For example, how to use DistFID in the proposed method should be introduced clearly. Why use dropout to replace random Gaussian noises as input?

[Response] Thank you for the comments and we have made the following modifications:

The definition of DistFID has been described in the Method section. We have added the implementation details in the “Distributed FID for image quality measurement” subsection. “At the beginning of training DSL, each client center sends the feature-wise statistics to the central center. Then, the central center can use the synthetic images and compute the DistFID value based on Equation 10 to evaluate the generator.”

Because the goal of the proposed framework is to produce paired images and labels for downstream applications (segmentation tasks in our study), we adopted the pix2pix method [50] and used the mask as the input condition to guide the generation of the synthetic image. Thus, our model does not take random Gaussian noises as input and uses dropout layers to provide randomness. We noticed a recent research, DatasetGAN [56], could produce image and label pairs from noise input, but it was built on top of a pre-trained StyleGAN which requires a very large number of training samples. We have added in future work: “Additional controlling factors for the generator (instead of segmentation masks), new techniques to generate both image and mask [56], and different downstream tasks can be further explored to assess the performance of DSL.”

5. In terms of multi-modal dataset, building multiple discriminators will increase a large or overtaken computation overhead. Is there any way to alleviate such burden? In addition, some

connections between different discriminators of one participant should be build up as they all come from one dataset.

[Response] Thank you. We have added more discussion in the Discussion section:

“In the multi-modality experiment, we built the MM-DSL with multiple discriminators that leads to superior performance compared to using one discriminator with multiple-channel input at each client. This design is generalizable and can handle the missing-modality scenario, so that it does not need to learn from a channel of empty modality data. The semantic correlations between different modalities are captured in the generator, which can synthesize multi-modal data as a multi-channel output. Thus, we did not introduce additional computations on the discriminator side to explicitly regularize the inter-modality connections.”

Overall, the proposed work has some contributions with noteworthy results, but the authors may revise some parts of the manuscript, and the reviewer would like to check the changes if they are well addressed.

[Response] Thank you very much. We have made detailed responses and added necessary discussions in the revised manuscript.

Reviewer #2 (Remarks to the Author):

This study proposed a distributed synthetic learning (DSL) architecture to address patient privacy and heterogeneity concerns in multi-center data scenarios. GAN-based synthetic learning is used as a main technique to shield the center information. The architecture contains one central generator and multiple node discriminators. The evaluation is conducted on three segmentation applications.

(1) Since DSL is designed to address medical privacy and heterogeneity challenges from multi-center datasets, it should be explained clearer how the heterogeneity affects the data and the model performance. At least, a sample of difference among multiple centers should be displayed.

[Response] Thanks for the suggestions and we modified the manuscript as follows:

In Table 1, we summarized the properties of the datasets used in our study. We have added sentences in “Data collection and processing” section to explain the data heterogeneity. “The data heterogeneity lies in several aspects, including the number of samples, acquisition scanners, resolutions, geographic locations, modality (the missing-modality setting), and organs (the histopathology dataset). Supplementary Figure 6, 7, and 8 show differences of some data samples among multiple centers.”

The following are the supplementary Figure 6, 7, and 8. We can see that the data in different sites have varied resolutions, image quality, appearance, and histogram distribution.

Figure 6: Cardiac CT image samples (6 samples per center) and histogram charts from three different datasets.

Figure 7: BraTS image samples (2 samples per center) and histogram charts from three different datasets.

Figure 8: Nuclei pathological image samples (6 samples per center) and histogram charts from four different datasets.

We also reported the segmentation results when only using the training samples from one of the centers in Table 2 (Real-WHS, Real-CAT08, Real-ASOCA), Table 3 (Real-CBICA, Real-TCIA, Real-Other). When using single-center data, the segmentation performance dropped due to data heterogeneity. We also validated a learning scenario when various modalities were available in multi-center datasets in Table 4. The Hetero-Real-FedSeg method directly learns a segmentation model from multi-center data with various missing modalities, while our

Hetero-MM-DSL learns a generator from the same multi-center missing-modality data and generate complete-modality data for training segmentation model. We can see that our method can produce significantly better results than Hetero-FedSeg which can not handle the missing modalities well.

(2) Only experiments for segmentation are reported, to fully evaluate the ability of DSL, it would be better to report experiments on some other types of applications, e.g., image reconstruction or classification.

[Response] Thanks for the suggestion. We have discussed in the Discussion section some possible future works: “Additional controlling factors for the generator (instead of segmentation masks), new techniques to generate both image and mask [56], and different downstream tasks can be further explored to assess the performance of DSL. For example, we can explore the use of bounding boxes or global labels to generate data for detection or classification, or even text-to-image generation [57].”

We agree that applying our method to different types of applications is important. Thus, we trained a downstream classification model using the BraTS synthetic dataset to recognize whether a tumor exists in the image. The classifier was trained on synthetic data and tested on real data. The results were added in the Results section and Table 5 of the revision.

Table 5: Multi-modality brain MRI binary classification task results. In the first column, 'Real-' indicates the model trained from original real images, otherwise, the model is trained from synthetic images. Real-All merges together all data of CBICA, TCIA, and *Other*. The classification model trained from MM-DSL’s synthetic data outperforms the other generative methods and is comparable to the models trained from real data.

Data/Method	Accuracy \uparrow	Sensitivity \uparrow	Specificity \uparrow
Real-All	0.923	0.895	0.944
Real-CBICA	0.874	0.798	0.929
Real-TCIA	0.89	0.855	0.916
Real-Other	0.743	0.681	0.787
FLGAN	0.653	0.932	0.460
FedMed-GAN	0.745	0.885	0.644
AsynDGAN	0.84	0.783	0.881
MM-DSL	0.897	0.853	0.928

(3) It seems that all the models (e.g., synthesis and segmentation models) are implemented in 2D convolutional networks, but some of the data (e.g., BraTS18) are 3D images. It would be better to conduct experiments on 3D networks for these data.

[Response] Thank you. We implemented our method using 2D networks purposely due to three reasons. First, as Reviewer 3 mentioned, a previous study [49] showed that 3D cranial MRI scans can be used to reconstruct facial images and leak a subject's identification, which could result in an infringement of privacy. To better protect privacy, we implemented our method in 2D which is robust to membership inference attacks as discussed in the responses to Reviewer1 Q3 and Reviewer3 Q1. Second, the number of available samples would be significantly smaller if we implement 3D models. It is well known that 3D networks need more data to train otherwise they are more prone to overfitting. For a GAN model, a limited dataset could easily cause model collapse. Third, a 3D implementation would require much larger GPU memories and increase the costs of computation and communication.

(4) In Figure 3, it is not professional to display brain images in that direction, please display them like that in Figure 1C.

[Response] Thanks for the comment. We have adjusted brain images in Figure 3 and Figure 4 in the revision as shown below.

Figure 3: Synthetic image examples and segmentation results for the multi-modality MRI brain experiment. (A) One example of real multi-modality MRI brain images (a), and the corresponding synthetic images generated by different methods with image quality metrics (b-e). (B) The Dist-FID and FID score curves over the training epochs. (C) Three examples of segmentation results for different methods vs the ground truth label (b). The segmentation model learned from MM-DSL’s synthetic data obtains more accurate results (j) than other methods (d-i) and is comparable to centralized learning (c).

Figure 4: Synthetic image examples and segmentation results for the missing-modality MRI brain experiment. (A) Three examples of real brain images from three data centers respectively (a, c, e), and the corresponding synthetic images from our method (b, d, f). The red dash boxes indicate the missing modality and the yellow solid boxes indicate the completed synthetic image generated by DSL. (B) Two examples of segmentation results for different methods vs the ground truth label (b). The segmentation model learned from Hetero-MM-DSL's synthetic data obtains more accurate results (f) than Hetero-Real-FedSeg (d) and does not have a clear performance drop compared to the complete-modality model (e).

(5) For Missing-modality Completion, BraTS18 seems is a dataset that contains complete modalities for all subjects. Why not consider other real incomplete datasets, such as ADNI, AIBL, A4? (<https://ida.loni.usc.edu/login.jsp?project=ADNI>)

[Response] Thanks for the comment.

We chose BraTS18 to evaluate the missing-modality experiment for two reasons. First, we can evaluate the effect of missing-modality completion by comparing the synthetic images with the corresponding real images provided by BraTS18. Other incomplete datasets do not have this support. Second, “It is worth noting that our framework is not specifically designed to complete the missing modality in the real data domain [51]. The missing-modality completion in our study happens in the synthetic image domain and makes the complete-modality synthetic database statistically appropriate for downstream semantic segmentation tasks. As shown in Fig.4(A), the synthetic image does not match pixel-to-pixel the real image, since the input image does not provide constraints at every pixel.”

We have added these sentences in the Discussion section for clarification.

(6) There are some typos, e.g., resnet->ResNet, DICE -> Dice, federated GAN model (FLGAN)->federated-learning GAN. The Dice score of DSL is 0.829 is remarkably -> The Dice score of DSL is 0.829 which is remarkably. Full spell of TDGAN is missing.

[Response] Thanks for the comment. We have revised the paper accordingly and corrected other typos and mistakes.

(7) As only segmentation is evaluated, it may be over-claimed that “The proposed framework demonstrates its potential for integrating multi-center heterogeneous data to support downstream clinical decision making.”

[Response] Thanks for the comment. We have revised the sentence as “The proposed framework is general and can be extended to support many other downstream machine-learning tasks.” and moved to Discussion section for the future work.

Reviewer #3 (Remarks to the Author):

This paper reports on the development and evaluation of a federated learning approach, namely distributed synthetic learning (DSL), to address the challenges of multi-party computation on privacy, multi-modal data, and dynamic collaboration issues. The approach is built on the models that the authors published previously and extends to solve the multi-modality simulation with possible missing modalities and temporal online/offline dynamics of participant data centers. The authors conducted an extensive set of evaluations and comparisons on three categories of medical image data sources, and the developed approach demonstrates better performance in generating high-quality image data and supporting the segmentation tasks compared to baseline methods. This paper is overall straightforward and easy to follow.

However, this reviewer has multiple major concerns to enumerate as follows.

1 This study focuses on a new federated learning paradigm that shares synthetic data and image masks (derived from each real image) between a central entity and multiple distributed entities, instead of sharing parameters and gradients as typical federated learning algorithms do. This creates a new scenario for privacy leakage/attacks, which was not involved/discussed in the manuscript. The authors claim that sharing synthetic data and image masks can secure privacy (“without leaking sensitive personal information” in Line 17, “provide a privacy-secured paradigm” Line 200), but it needs to be investigated before making such a conclusion. Since the GAN training is dictated by the shared image masks from distributed entities, the generated “fake” images look very similar to their paired real images from distributed entities (visual evidence can be found in Fig 2A and 3A, where each synthetic image ample is almost the same as the corresponding real image). That means, even though a private entity receives a “synthetic dataset” generated in a centralized manner, each image of the dataset still reserves a link to a real person in another entity with high confidence. In an extreme case, sharing such a synthetic dataset with all private entities is similar to sharing real images directly. This creates concerns about the potential of privacy breaches, such as reidentification and membership attacks. It is also known that patients can be reidentified through reconstructing faces from MRI data.

[1] Schwarz CG, Kremers WK, Therneau TM, Sharp RR, Gunter JL, Vemuri P, Arani A, Spsychalla AJ, Kantarci K, Knopman DS, Petersen RC. Identification of anonymous MRI research participants with face-recognition software. *New England Journal of Medicine*. 2019 Oct 24;381(17):1684-6.

[2] der Goten V, Alexander L, Hepp T, Akata Z, Smith K. Conditional De-Identification of 3D Magnetic Resonance Images. *arXiv preprint arXiv:2110.09927*. 2021 Oct 18.

[Response] Thanks for the insightful comments. We have added detailed and necessary discussions below in the Results and Discussion sections about possible privacy attacks.

In the Results section, we added the following paragraphs:

“Membership Inference Attack Evaluation The ability to defend against malicious attacks is vital for building privacy-preserving machine-learning applications in medicine, especially in a distributed learning scenario. To validate the robustness of DSL under potential adversarial attacks, we extend to perform key DSL ablation studies. We focus on the membership inference attack which is a type of adversarial attack for machine learning models [35] that relates to privacy concerns. In our study, given a real data record (images), and black-box access to the trained image generator (setting 1) or the synthetic database (setting 2), an attacker would like to determine if the real record was in the model's training dataset. We analyzed two important settings for the membership adversarial attack using the BraTS dataset.

In the first attack setting, only the trained generator is accessible as an API to attackers. An attacker can obtain the synthetic image given an input mask and compare it with the

corresponding real image. The hypothesis is that a trained sample is more similar to its corresponding synthetic image than an unseen sample, and thus one attacker can utilize the similarity between the paired real and synthetic images to achieve a membership inference attack. We adopted two image similarity metrics, including the cosine distance score of the perceptual features from pretrained resnet50 model on ImageNet (perceptual similarity = 1 - cosine distance) and the structural similarity index measure (SSIM). The statistics of perceptual similarity for 11,349 trained samples vs 2,730 unseen samples are 0.996962 ± 0.000195 vs 0.996963 ± 0.000194 ($p=0.8946$). The statistics of SSIM for the trained samples vs unseen samples are 0.803466 ± 0.049409 vs 0.804076 ± 0.048799 ($p=0.5615$). From these details, there is no statistical difference between trained samples and unseen samples in terms of these similarity metrics. Alternatively, a recent study proposed an effective membership inference attack technique [36] that worked on all major cloud-based machine learning services. The method first learned multiple "shadow models" that imitate the behavior of the target mode and then learned an attack model based on the outputs of the shadow models [36]. However, in our use case, the attacker can hardly learn good "shadow models" because of no access or knowledge of the discriminators and difficult training of GANs in practice [37]. It is therefore recognized that an effective membership attack is not feasible for our DSL framework.

In the second attack setting, only the transformed synthetic database is accessible to attackers. An attacker can use image retrieval techniques to search for the most similar match for a given real image, and then infer the membership based on the image similarity. In this second setting, we used a fast (approximated) nearest neighbor method <https://github.com/pixelogik/NearPy> and the perceptual features from pretrained resnet50 to implement the image retrieval. We assume that the attacker could access the other site (975 training samples) in the brain task and another 975 unseen samples to train an attack model (classifier) based on two features, including the perceptual similarity and SSIM between the real image and the corresponding retrieved synthetic image. The attack model was then tested on randomly selected 1,750 training samples from the TCIA and CBICA sites and 1,750 unseen samples. The classification results have accuracy 0.5 and F1 score 0.5 using a linear support vector classification (SVC), accuracy 0.47 and F1 score 0.52 by an RBF-kernel SVC [38]. We see that the synthetic database with random transformations demonstrates its robustness to defend the membership inference attacks. The privacy preservation in our framework could be further enhanced by incorporating additional security mechanisms like differential privacy [39] (all federated-learning methods in our experiments were trained without differential privacy). However, introducing differential privacy comes with a notable accuracy performance cost [40]. Introducing differential privacy to MM-DSL in the BraTS experiment makes the segmentation accuracy drop from Dice 0.829 ± 0.128 in Table 3 to 0.730 ± 0.201 ."

We also added the following paragraphs in the Discussion section:

"DSL ensures the security of private clinical data for the following two reasons. First, during model training, DSL transmits only synthetic images and the corresponding losses which prevents the transmission of real DICOM images with sensitive patient information [47]. In the

standard federated multi-node training where the same model is shared, an adversarial attack [48] can recover the original image data from model gradients. Such an attack does not work on DSL, because DSL does not have any model shared in any data-center. The central server and all participants learn different models and no model parameters are exchanged.

Second, during the synthetic data generation, random transformations (e.g., scaling, rotation, shift) can be applied to the input. This operation introduces additional image variances beyond the random dropout operations, providing an extra layer of data security protection. A key advance is that the randomly transformed synthetic dataset can lead to similar downstream segmentation performance, while avoiding generation of close-to-identical training samples. For example, in the brain tumor experiment, the segmentation metrics for training on the transformed synthetic database are Dice 0.838 ± 0.121 , HD95 14.68 ± 15.74 , SD 2.26 ± 1.93 , which are close to the MM-DSL results in Table 3. Note that all GAN-based methods being compared in Table 3 used not augmented synthetic data.

We also noticed that a previous study [49] showed the possibility of identification of anonymous 3D cranial MRI scans using face-recognition technology. However, our method was implemented using 2D networks and 2D images with privacy preservation. In our study, 2D medical images were indexed randomly and used independently without the original 3D information. Thus the generator does not learn any 3D information explicitly. Additionally, the random transforms in the generation stage makes 3D facial reconstruction impossible. Fig. 2(A) and Fig. 3(A) show realistic-looking synthetic image examples and demonstrate the ability of our method to generate high-quality images. Meanwhile, Fig. 4(A) shows that in unconditioned areas the synthetic image can have very different semantic appearances (e.g., smaller brain ventricles) compared with the real image data. In summary, the generated synthetic database does makes it impossible the 3D face reconstruction of a patient.”

2 Since the diversity of synthetic data is largely determined by the available image masks, which are derived from real images, the variance of generated images by the trained GAN model seems very limited. The premise of generating synthetic images is to obtain a decent number of image masks. However, this really limits the ability of synthetic data generation. This study does not provide solutions on how to generate diverse datasets (or how to obtain image masks) that do not rely upon the available image masks.

[Response] Thanks for the valuable comments. We have discussed synthetic data augmentation in the Discussion section.

First, we aim to propose a distributed synthetic learning framework that can enable effective collaborative learning between multiple data centers without sharing real data. In order to produce meaningful synthetic image-mask pairs for downstream segmentation purposes, we borrowed the idea of image-to-image translation [50] and utilized mask as the input condition to guide the generation process. “This randomness does not offer significant variations to augment the synthetic data, thus we have applied separate data augmentation methods when generating a synthetic dataset for the downstream segmentation task. For instance, we applied random

transformations to the input, such as scaling, shifting, flipping, and rotation, to generate more diverse images (see examples of synthetic images in Supplementary Fig 3, 4, 5). Also, for datasets including multi-labels like the BraTS dataset, we can adopt a mixture of different brain skulls with tumor regions, introducing additional variations in the labels and synthetic datasets. This key operation differentiates the synthetic samples from the real samples while preserving the distribution of the real data.”

“Second, the size of the original data affects the performance gain of the data augmentation method. When the size of all real data is small, scaling up the synthetic database by random transformations brings the benefit of increased variance in the synthetic database. We conducted an ablation study by using the histopathology data in a 4 data-center distributed learning setting similar to the cardiac and brain tasks. By generating a synthetic dataset twice the size of the real data, the augmented dataset used improves the segmentation performance (Dice: from 0.789 to 0.805, AJI: from 0.528 to 0.552). However, further increasing the size of the synthetic database does not benefit the downstream models.”

Third, we agree that generating diverse labeled datasets (obtain both images and masks) that do not rely upon the limited real data is of interest and still an open challenge. We have included the idea into the future work in the Discussion section. “Additional controlling factors for the generator (instead of segmentation masks), new techniques to generate both image and mask [56], and different downstream tasks can be further explored to assess the performance of DSL. For example, we can explore the use of bounding boxes or global labels to generate data for detection or classification, or even text-to-image generation [57].”

3 It is unclear what are the differences between the major baseline model (ie, AsynDGAN, the authors’ CVPR paper) and the proposed DSL. In evaluating model performance on the Cardiac CTA datasets (single modality), the authors state that DSL employs AsynDGAN’s structure (Line 241), but DSL and AsynDGAN show different performances in Table 1. This is confusing and needs clarification. Also, subsection “Extension for temporal datasets” is very confusing because the authors mainly introduce the optimization objective of a baseline model, instead of the proposed model. Their relationship should be clearly stated.

[Response] Thanks for the comment. This new submission (DSL) is significantly different from the baseline [27] in the following aspects:

1. DSL extends to support multi-modality data learning by introducing key changes to the network architecture, i.e., multiple discriminators at each data center. As a result, the new method of DSL can learn to generate high-quality multi-modality medical images. Furthermore, this novel design enables the method to learn from missing-modality data.
2. AsynDGAN in [27] was validated by evenly split data sets. In this study, we adopt more practical settings and separate samples based on the original center information to show that the new method DSL can learn from heterogeneous data, e.g., different size of data sets and various imaging protocols. To this end, a weighted aggregation strategy has been adopted in DSL to handle varied data sizes.
3. A new metric for image quality, named dist-FID, is proposed to track the synthetic image quality without direct access to the original real images. The new metric enables automatic model selection after training. In the baseline [27], the best model was

selected by visual inspection of the synthetic examples. This is also the main difference between the CL-DSL and the baseline TDGAN method [34] in the continuous learning experiment.

We have added sentences to describe the baseline method: “AsynDGAN is a baseline distributed GAN model that showed promising performance in learning to synthesize T2 brain images and histopathology images. It shares a similar architecture as DSL, but lacks support for heterogeneous data and an efficient model selection strategy.”

4 The authors report comparison results between CL-DSL and FLGAN (Line 179) but it is hard to figure out where these numbers and comparisons come from. According to the stated experimental setting in Lines 169-175, it seems that such comparisons were not planned/described. This makes the results in this paragraph (especially Lines 189-191) hard to follow.

[Response] Thanks for the comments. We apologize for the misleading information in the last paragraph of the Results section. The comparisons between DSL and FLGAN were not for the setting of continuous learning, thus, we have carefully revised this section by removing those sentences. We also removed a “Local-GAN” setting from the experiment because it is not a continual-learning setting.

5 This reviewer cannot access the source code for evaluation because the provided link (<https://rutgers.app.box.com/folder/147492331099>) is private.

[Response] Thanks for the comment. We have updated the link (https://github.com/tommy-qichang/DSL_All_Code) and made sure the source code is accessible. We also included demo scripts to show the learning of DSL and the segmentation models.

6 It is unclear what are the key takeaways from Figures 2C, 3C, 4B, and 5C.

[Response] The Figures 2C, 3C, 4B, and 5C show some examples of segmentation results from different methods in a qualitative way. We can see by comparing with the ground truth mask images that the segmentation models trained from the synthetic images which are generated by the proposed method can obtain consistently better segmentation results than the models using synthetic images from other generative methods. We have added short comments in the figure captions. For example, in Fig. 2, we added “The segmentation model learned from DSL's synthetic data obtains more accurate results (j) than the other methods (d-i) and is comparable to centralized learning (c).”

7 This reviewer cannot understand the undefined term $G_t(u,y)$ in Eq 8. Why a generator can be fed with a real image y and some noise? The whole Eq 8 was copied from the authors' previous publication but there exist definition inconsistencies, making it hard to follow.

[Response] Thanks for the comments. We have revised the notations in Eq 8 to make the notations consistent in the paper.

$$\begin{aligned}
V_t(G_t, D_t^{1:K_t}) &= \min_{G_t} L_{Digesting} + \lambda \cdot L_{Reminding} \\
\text{Digesting Loss : } L_{Digesting} &\triangleq \max_{D_t^{1:K_t}} \sum_{j=1}^{K_t} \pi_t^j \mathbb{E}_{x \sim s_t^j(x)} \left\{ \mathbb{E}_{y \sim p_{data}(y|x)} [\log D_t^j(y|x)] \right. \\
&\quad \left. + \mathbb{E}_{\hat{y}_j \sim p_{\hat{y}}(\hat{y}_j|x)} [\log(1 - D_t^j(G_t(x)|x))] \right\} \\
\text{Reminding Loss : } L_{Reminding} &\triangleq \mathbb{E}_{x \sim s_{t-1}(x)} \mathbb{E}_{\hat{y}_j \sim p_{\hat{y}}(\hat{y}_j|x)} [\|G_t(x) - G_{t-1}(x)\|^2]
\end{aligned} \tag{8}$$

This reviewer also has several specific concerns.

1 The authors select FedSeg and Hetero-FedSeg as two major baseline methods for Hetero-MM-DSL, but neither references nor descriptions were provided for review. The authors also need to provide justifications on why FedSeg was selected as a baseline. It is also unclear whether Hetero-FedSeg uses synthetic images to learn the model or not. If not, it would be better to name it something like Hetero-Real-FedSeg.

[Response]

Thanks for the comments. We have renamed the 'Hetero-FedSeg' to 'Hetero-Real-FedSeg' to make it clear that the model was learned from missing-modality real data. We added the following descriptions in the Results section to explain FedSeg and Hetero-Real-FedSeg.

"We compare Hetero-MM-DSL with the federated segmentation method FedSeg {<https://fedsegment.github.io/home/>} which learns directly from multi-channel real data. Real-FedSeg represents the FedSeg when it learns from complete-modality real data, while Hetero-Real-FedSeg is FedSeg when it learns from missing-modality data. Each missing modality was represented as a channel of all zeros. We selected FedSeg to compare with our DSL for the following two major reasons. First, FedSeg is a federated learning method that can learn from distributed data. Second, in this case directly learning a segmentation model from real data is considered an upper bound of learning from synthetic data. Since the MM-DSL already achieved better results than the other GAN-based methods as shown in Table 2 and Table 3, we just need to compare the results of Hetero-MM-DSL with the results of Hetero-Real-FedSeg and present them in Table 4."

2 In the caption of Table 5, the authors mentioned "Real- indicates the model ..."; however, there is no model with the prefix "Real-" in Table 5.

[Response] Thanks for the detailed comment. We have corrected the error and revised the caption of Table 6 (the index was updated in the revision).

Table 6: Quantitative nuclear segmentation results (average Dice and AJI) for the continual learning setting. All comparing methods use the GAN model and the segmentation models are trained from corresponding synthetic data. Note that the results in the Task1 columns are the same among all the methods because the learning processes of Task1 are the same for different approaches. CL-DSL outperforms the other methods and obtains close performance to the centralized-learning JCL.

Method	Task1 (liver)		Task2 (breast)		Task3 (kidney)		Task4 (prostate)	
	Dice ↑	AJI ↑	Dice ↑	AJI ↑	Dice ↑	AJI ↑	Dice ↑	AJI ↑
JCL	0.6676	0.3420	0.7114	0.4457	0.7350	0.4814	0.7627	0.5184
Fine-Tuning	0.6676	0.3420	0.6950	0.4405	0.7142	0.4195	0.6902	0.4273
TDGAN	0.6676	0.3420	0.6961	0.4323	0.7164	0.4512	0.7481	0.4931
CL-DSL	0.6676	0.3420	0.7346	0.4638	0.7428	0.4605	0.7633	0.4828

3 Redundancy needs to be removed for subsection “Data collection and processing”. Many details have been introduced in the Data subsection. The authors can probably merge these two subsections.

[Response] Thanks for the detailed comment. We have revised the paper by merging the ‘Data’ and ‘Data collection and processing’ subsections.

4 The authors claim (Line 291) that the BraTS18 dataset was split to 170 and 40 for training and testing; however, these seem inconsistent with the numbers in Table 1 (168 vs 42).

[Response] Thanks for the detailed comment. We have corrected the error. The correct numbers are 168 vs 42.

5 The authors use “MM-DSL” to refer to the DSL architecture that considers the setting of multi-modality images (Line 127); however, the authors never use this name in Results and Methods, but used “Hetero-MM-DSL” for the missing modality architecture. It seems that “DSL” in Tables 3 and 4 should be “MM-DSL”, and all “DSL” refer to the full modality architecture should be renamed accordingly.

[Response] Thanks for the detailed comment. We have revised the text and Tables to use ‘MM-DSL’ for multi-modality DSL, ‘Hetero-MM-DSL’ for the missing-modality DSL.

6 Cell “40 (6360)ss” in Table 1 needs to be corrected.

[Response] Thanks for the detailed comment. We have corrected the error by removing ‘ss’ and also updated several incorrect numbers in Table 1.

7 Function $G_s(x)$ not defined before use (Line 405).

[Response] Thanks for the detailed comment. We have revised the sentence: “we can use the synthetic image at the s 'th channel of $G(x)$ as a substitute.”

8 “t_4” in Line 168 seems unnecessary and can be removed.

[Response] Thanks for the detailed comment. We have removed this.

9 There are typos. For example, Line 230 “higher qualify” -- > “higher quality”

[Response] Thanks for the detailed comment. We have corrected the errors throughout the paper.

Reviewer comments, second round

Reviewer #1 (Remarks to the Author):

Thanks for the detailed response. Basically, the authors have provided a reasonable reply to the previous concerns. Some minor comments are listed below.

1. The authors should polish the manuscript as some typos exist in the current version. For example, "resnet" should be "ResNet".
2. The provided figure in the reply letter is not clear enough. Some modifications can be taken for a better review.
3. In terms of comment 5, the authors said "we extend our framework by deploying multiple discriminators at each entity" on page 14, thus the reviewer wonders if some computation overhead will happen in one entity. The replies seem to be unclear about this. The authors are suggested to provide a more detailed explanation.

Reviewer #3 (Remarks to the Author):

This reviewer appreciates the authors for their deep revisions of the manuscript in the last round. The authors have addressed the majority of concerns/suggestions of this reviewer and the quality of this study has been vertically improved. However, this reviewer has further concerns as follows.

1. The comparisons in the caption of Fig.2C are based on eyeball checks, which is difficult for readers to follow. Please provide related metric values and mark them to the figure for easy reading.
2. In the evaluation of the membership inference risk of DSL, this reviewer doesn't think that the way of treating the nearest neighbor synthetic image as "positive" and all others as "negative" is reasonable. In membership disclosure literature [1], researchers use a threshold of the similarity between real and synthetic data to determine the correctness of an inference. Although there is no clear guidance on the threshold selection for this scenario, this method is still close to attack practice. For example, if all available synthetic data are far away from a given real data, then the attacker will not make any positive inference.

The authors need to be careful in using similarity metrics, such as SSIM, in this task as it cannot deal with even minor image transformation (rotation, scaling, ...). For example, in supplemental Figure 3, human eyeball check can figure out very easily that in each column the lower 3 generated fake images are from the upper real data in the same column, and NOT from the real data in other columns. The same observation applies to supplemental Figure 4. These correspond to successful membership inference attacks even though there are some minor differences in position, scaling, edge, etc. The result of $F1=accuracy=0.5$ for the trained attack model in this case does not seem reasonable.

3. A side message is that in the medical domain, the second scenario of the membership inference attack is much more realistic than the first one because sharing generators can pose greater risks than just releasing data or post-processed data.

[1] Yan C, Yan Y, Wan Z, Zhang Z, Omberg L, Guinney J, Mooney SD, Malin BA. A Multifaceted benchmarking of synthetic electronic health record generation models. *Nature Communications*. 2022 Dec 9;13(1):7609.

Response to reviewers' comments for NCOMMS-22-35277A

We thank the reviewers for their additional comments on our revised manuscript, NCOMMS-22-35277A. We have addressed all comments raised by the reviewers. The modifications we made are highlighted in blue within the updated revision. Below, we provide detailed responses to each reviewer's comments. Please note that all table indices, section indices, reference indexes, and page numbers mentioned refer to the revised manuscript, NCOMMS-22-35277B.

Reviewer#1:

1. The authors should polish the manuscript as some typos exist in the current version. For example, "resnet" should be "ResNet".

[Response] Thank you for your comment. We have conducted a thorough proofreading of the manuscript, and we have made all necessary corrections and updates.

2. The provided figure in the reply letter is not clear enough. Some modifications can be taken for a better review.

[Response] We value the reviewer's feedback. The figures included in the response letter for revision 1 may not have been sufficiently clear since they were screenshots taken directly from the manuscript's PDF file to highlight the relevant modifications. However, in this second revision, we have ensured to incorporate the original high-resolution figures in the response letter to enhance the review process.

3. In terms of comment 5, the authors said "we extend our framework by deploying multiple discriminators at each entity" on page 14, thus the reviewer wonders if some computation overhead will happen in one entity. The replies seem to be unclear about this. The authors are suggested to provide a more detailed explanation.

[Response] Thanks for the suggestion. We have provided a clear and detailed explanation in the Discussion section as follows:

"In the multi-modality experiment, we constructed the MM-DSL with multiple discriminators at each center, where each discriminator focuses on a specific modality. While this design necessitates training using more parameters at each data center compared to using a single discriminator with multi-channel input for multi-modality data, it results in a superior generative model. Specifically, there is an improvement in dist-FID from 29.37 to 28.06, and downstream segmentation Dice from 0.821 ± 0.16 to 0.829 ± 0.128 . This improvement is accompanied by a slight increase in overall training time (from 37.6 minutes per epoch to 40 minutes per epoch) and GPU memory usage for each data center (from 6.5 GB to 6.9 GB). Furthermore, our framework eliminates the requirement for training the generator at each data center, resulting in significant computational savings compared to other approaches such as FLGAN and FedMed-GAN."

Reviewer#3:

1. The comparisons in the caption of Fig.2C are based on eyeball checks, which is difficult for readers to follow. Please provide related metric values and mark them to the figure for easy reading.

[Response] We appreciate the feedback. In the revised manuscript, we have provided related Dice scores and marked them in Figs. 2-5 to improve the readability.

Figure 2

Figure 3

A

B

Figure 4

generated fake images are from the upper real data in the same column, and NOT from the real data in other columns. The same observation applies to supplemental Figure 4. These correspond to successful membership inference attacks even though there are some minor differences in position, scaling, edge, etc. The result of $F1=accuracy=0.5$ for the trained attack model in this case does not seem reasonable.

[1] Yan C, Yan Y, Wan Z, Zhang Z, Omberg L, Guinney J, Mooney SD, Malin BA. A Multifaceted benchmarking of synthetic electronic health record generation models. *Nature Communications*. 2022 Dec 9;13(1):7609.

[Response] Thank you for the valuable suggestions and comments. We have included the reviewer's reference [1] in our revision. In our response and revision, we use [36], which corresponds to the reviewer's reference [1].

First, we would like to clarify the evaluation of the membership inference risk of DSL. Our evaluation follows a similar approach to the analysis of membership inference risk described in reference [36]. In [36], a target record was classified as positive (used to train the GAN), if the Euclidean distance between this record and any one of the synthetic records in terms of all the attributes was smaller than an empirically hard-coded threshold. Essentially, their algorithm is equivalent to first determining the closest synthetic record to the target and then comparing the distance with the threshold. Unlike electronic health records, extracting a proper representation of image for a specific task is not straightforward. Therefore in our evaluation, we need to derive some proper image attributes and train a classifier instead of defining a hard-coded threshold. In our previous revision, we first search the closest synthetic image to the target using image perceptual feature, which is extracted from a pretrained ResNet50 model on the ImageNet database. The perceptual distance is defined as $1 - \text{cosine}$ of two image features. We adopt a hash-based image retrieval technique instead of a simple traversal algorithm due to its computational efficiency. In the classification step, we use two image similarity metrics to represent a real image and train an SVM classifier. The metrics include the perceptual distance and the SSIM between the real image and its closest synthetic image.

Since the discussion of image retrieval technique is beyond the scope of our study, we adopt the traversal search like in [36] instead of the image retrieval for the closest synthetic image to a real image in this revision. Based on the reviewer's comments, we replace the SSIM with the normalized-root-mean-squared error as another image similarity metric beside the perceptual distance. In the traversal step, we normalize these two similarity distances into a range of zero and one and use their sum to search for the closest synthetic image. In the classification step, these two distances between a real image and its closest synthetic image are used as two independent features of the real image. We also update the settings in both scenarios for a more realistic evaluation.

Note that the major reason why human eyeball check can easily match the generated images to the corresponding real data in supplemental Figs 3 and 4, is the different anatomical contextual information between different samples. We have found two challenges to automatically and correctly identify the synthetic image from a large synthetic image dataset that corresponds to the real image. Firstly, the 2D medical images from the same anatomical positions usually have

similar contextual information across different patients which can lead to many ambiguities. Secondly, the affine transformations (such as rotation, scaling) and pixel-wise intensity changes in the synthetic images make it harder to find the correct match because there is no ideal similarity measurement to tackle all these challenges and the currently well-known metrics do not work well.

We have made the following updates as seen in the revision:

“In our study, we consider two types of attack settings to analyze the membership inference risk on the BraTS dataset. In setting 1, the attacker has access to a set of real images and the transformation-augmented synthetic database. Specifically, the attacker can access a set of real images from the *Other* center, including 300 images used for training the DSL (positive samples) and 300 images not used for training (negative samples). We evaluate the attacker's performance on a randomly selected set of real images from the CBICA and TCIA centers, consisting of 1000 positive and 1000 negative samples. We adopt a membership inference risk analysis similar to [36]. First, we calculate image similarity metrics between each real image and each synthetic image using the normalized root-mean-square error and perceptual distance. The perceptual distance is defined as $1 - \cos(\text{images' perceptual features})$, which are extracted from a pretrained ResNet50 model on the ImageNet database. These two metrics are normalized and summed to identify the closest synthetic image to each real image. Then, we use the two similarity metrics from the closest synthetic image as independent features to represent the real image and train SVM classifiers [37] on the 600 samples, and test on the 2000 samples. The linear support vector classifier (SVC) exhibits low testing accuracy, with an F1 score of 0.58, recall of 0.52, precision of 0.66, and AUC of 0.65. Similarly, the performance of the RBF-kernel SVC is also low, with an F1 score of 0.57, recall of 0.53, precision of 0.60, and AUC of 0.62. The membership inference attack faces two significant challenges. Firstly, 2D medical images from the same anatomical positions typically exhibit similar contextual information across different patients. This similarity can create numerous ambiguities during the attack. Secondly, the presence of affine transformations (such as rotation and scaling) and pixel-wise intensity differences in synthetic images make it more difficult to identify the correct match for the target. These challenges arise due to the absence of an ideal similarity measurement that can effectively address all these complexities.

In setting 2, the attacker has black-box access to the trained image generator through an API. By providing a real image's mask as input to the API, the attacker can obtain a corresponding synthetic image and compare it with the real image. We utilize two image similarity metrics (normalized root-mean-square error and perceptual distance) between paired real and synthetic images to perform a membership inference attack. Specifically, we use the same real data samples as in the previous setting to train and test SVM classifiers. By employing a linear support vector classifier (SVC), we obtain classification results on the testing samples, yielding an F1 score of 0.54, recall of 0.55, precision of 0.53, and AUC of 0.55. The utilization of an RBF-kernel SVC produces slightly lower results, with an F1 score of 0.45, recall of 0.38, precision of 0.54, and AUC of 0.55. The failed attack indicates that our trained generator has strong generalization and is not overfitting to the training data.”

3. A side message is that in the medical domain, the second scenario of the membership inference attack is much more realistic than the first one because sharing generators can pose greater risks than just releasing data or post-processed data.

[Response] Thank you for the valuable insight. In the "Membership inference risk evaluation" section, we have made a revision by swapping the order of the two attack scenarios based on the reviewer's comment. In this revised version, we first discuss the scenario where the synthetic dataset is released, as it is considered more realistic. Subsequently, we discuss the scenario where the trained generator is accessible as a black-box API. In this setting, the attacker can only get the synthetic image as output of the API, but does not have access to the generator model parameters. Our analysis covers both settings thoroughly.